# Goal-Conditioned Supervised Learning for Multi-Objective Recommendation

## Abstract

Multi-objective learning endeavors to concurrently optimize multiple objectives using a single model, aiming to achieve high and balanced performance across diverse objectives. However, this often entails a more complex optimization problem, particularly when navigating potential conflicts between objectives, leading to solutions with higher memory requirements and computational complexity. This paper introduces a Multi-Objective Goal-Conditioned Supervised Learning (MOGCSL) framework for automatically learning to achieve multiple objectives from offline sequential data. MOGCSL extends the conventional GCSL method to multi-objective scenarios by redefining goals from one-dimensional scalars to multi-dimensional vectors. It benefits from naturally eliminating the need for complex architectures and optimization constraints. Moreover, MOGCSL effectively filters out uninformative or noisy instances that fail to achieve desirable long-term rewards across multiple objectives. We also introduces a novel goal-selection algorithm for MOGCSL to model and identify "high" achievable goals for inference.

While MOGCSL is quite general, we focus on its application to the next action prediction problem in commercial-grade recommender systems. In this context, any viable solution needs to be reasonably scalable and also be robust to large amounts of noisy data that is characteristic of this application space. We show that MOGCSL performs admirably on both counts by extensive experiments on real-world recommendation datasets. Also, analysis and experiments are included to explain its strength in discounting the noisier portions of training data in recommender systems with multiple objectives. Code and data can be found in: https://anonymous.4open.science/r/MOGCSL-D7A2.

## 1 Introduction

Multi-objective learning techniques typically aim to train a single model to determine a policy for multiple objectives, which are often defined on different types of tasks Ruder (2017); Sener & Koltun (2018); Zhang & Yang (2021). For instance, two common tasks in recommender systems are to recommend items that users may click or purchase, and hence the corresponding two objectives are defined as pursuing higher click rate and higher purchase rate. However, learning for multiple objectives simultaneously is often nontrivial, particularly when there are potential inherent conflicts among these objectives Sener & Koltun (2018).

Existing approaches to multi-objective learning broadly address this optimization issue by formulating and optimizing a loss function that takes into account multiple objectives in a supervised learning paradigm. Some previous research focused on designing model architectures specifically for multi-objective scenarios, including the shared-bottom structure Ma et al. (2018), the experts and task gating networks Ma et al. (2018), and so on. Another line of research studies how to constrain the optimization process based on various assumptions regarding how to assign reasonable loss weights Liu et al. (2019) or dynamically adjust gradients Yu et al. (2020). We remark that these approaches often introduce substantial space and computational complexity Ma et al. (2018); Zhang & Yang (2021). Also, they treat all the data uniformly. In recommender systems, sometimes none of the items shown to a user is of interest, rendering their choice uninformative. Also users are often distracted or their interests temporarily change. These are some of the reasons that the observed interaction data in real settings suffers from substantial uninformative or noisy components that are better left discounted. Existing approaches do not cater well to this aspect of our focus application.

To address these issues, we propose a novel method called **Multi-Objective Goal-Conditioned Supervised Learning (MOGCSL)**. In this framework, we first apply goal-conditioned supervised learning (GCSL) Yang et al. (2022); Liu et al. (2022) to the multi-objective recommendation problem by introducing a new multi-dimensional goal. At its core, MOGCSL aims to learn primarily from the behaviors of those sessions where the long-term reward ends up being high, thereby discounting noisy user choices coming with low long-term rewards. Unlike conventional GCSL, however, we represent the reward gained from the environment with a vector instead of a scalar. Each dimension of this vector indicates the reward for a certain objective. The "goal" in MOGCSL can then be defined as a vector of desirable cumulative rewards on all of the objectives given the current state. By incorporating these goals as input, MOGCSL learns to rely primarily on high-fidelity portions of the data with less noise corresponding to multiple objectives, which helps to better predict users' real preference. Extensive experiments indicate that MOGCSL significantly outperforms other baselines and benefits from lower complexity.

For inference of GCSL, most previous works employ a simple goal-choosing strategy Chen et al. (2021b); Xin et al. (2022) (e.g., some multiple of the average goal in training). Although we observe that simple choices for MOGCSL do reasonably well in practice, we also introduce a goal-choosing algorithm that estimates the distribution of achievable goals using variational auto-encoders, and automates the selection of desirable "high" goals on multiple objectives as input for inference. This algorithm addresses the key challenge of selecting achievable "high" multi-dimensional goals, a problem that has remained unexplored yet. By comparing the proposed method with simpler statistical strategies, we gain valuable insights into the goal-choosing process and trade-offs for practical implementation.

Our key contributions can be summarized as follows:

- We introduce a general supervised framework called MOGCSL for multi-objective recommendations that, by design, selectively leverages training data with desirable long-term rewards on multiple objectives. We implement this approach using a transformer encoder optimized with a standard cross-entropy loss, avoiding more complex architectures or optimization constraints that are typical for multi-objective learning. Empirical experiments on real-world e-commerce datasets demonstrate the superior efficacy and efficiency of MOGCSL.

- As a part of MOGCSL, we conduct a formal analysis of goal properties for inference with a theorem. Then we introduce a novel goal-choosing algorithm that can model the distribution of achievable goals over interaction sequences and choose desirable "high" goals across multiple objectives. This is novel even for the single-objective case and addresses a fundamental challenge in GCSL.

- We conduct a comprehensive analysis of MOGCSL's working mechanism and are the first to reveal its ability to effectively mitigate harmful effects of noisy instances in the training data with multiple objectives, which is a crucial challenge in practical applications like recommender systems.

## 2 RELATED WORK

**Multi-Objective Learning.** Multi-objective learning typically investigates the construction and optimization of models that can simultaneously achieve multiple objectives. Existing research focuses mainly on resolving the problem by model architecture designs Ma et al. (2018); Misra et al. (2016) and optimization constraints Liu et al. (2019); Lin et al. (2019). All of these works give equal weight to all instances in the training data, instead of forcefully distinguishing noisy data from non-noisy data by considering their different effects on multiple objectives. This is fine in many applications but problematic in commercial recommendation systems. Moreover, these approaches often introduce substantial space and computational complexity Zhang & Yang (2021), making them more challenging for large-scale applications in the real world. For example, MMOE Ma et al. (2018) requires constructing separate towers for each objective. DWA Liu et al. (2019) necessitates recording and calculating loss change dynamics for each training epoch, while PE Lin et al. (2019) demands substantial computational resources to solve an optimization problem for Pareto efficiency. A recent work FAMO Liu et al. (2024) aims to improve the efficiency of multi-objective learning, but still suffers from the complexity of calculating gradient similarities to determine task weights.

**Goal-Conditioned Supervised Learning.** In contrast, we propose resolving the multi-objective optimization dilemma within the framework of GCSL Liu et al. (2022); Chen et al. (2021b); Janner

et al. (2021). Typically, GCSL can be directly combined with various sequential models with minor adaptations and trained entirely on offline data. This paradigm effectively transforms offline reinforcement learning into a supervised learning problem. However, as far as we know, most existing works focus on optimizing a single objective. Our work extends GCSL to the multi-objective setting, eliminating the need for scalarization functions or other constraints during training. Additionally, although some works have explored how to assign more valuable goals to enhance GCSL training Ajay et al. (2020); Zhuang et al. (2024), the properties and determination inference goals, especially multi-dimensional ones, remain less explored. We propose a novel algorithm to model the achievable goals and automatically choose desirable goals on multiple objectives as input during inference stage. Note that we are solving for next action prediction problem, and use long-term rewards only as extra information, unlike multi-objective reinforcement learning approaches that aim to maximize cumulative returns across multiple objectives Cai et al. (2022); Stamenkovic et al. (2022). Also, their evaluation principles are different, typically relying on long-term metrics and synthetic environments. Hence we don't compare with such approaches in this paper.

## 3 METHODOLOGY

In this section, we first illustrate the general optimization paradigm of MOGCSL. Then we expound on the training process of MOGCSL and the proposed goal-choosing algorithm for inference. Furthermore, we give a detailed analysis of the capability of MOGCSL to discount potentially highly noisy samples in the training data.

### 3.1 A NEW VIEW FOR MULTI-OBJECTIVE LEARNING

Multi-objective learning is typically formulated as an optimization problem over multiple losses Ma et al. (2018); Misra et al. (2016); Yu et al. (2020); Lin et al. (2019), each defined on a distinct objective. Consider a dataset $\mathcal{D} = \{(\mathbf{x}_i, y_i^1, y_i^2, ..., y_i^n)\}_{i \in [1,M]}$, where $\mathbf{x}_i$ represents the feature, $y_i^j$ is the ground-truth score on the $j$-th objective, and $M$ is the total number of data points. For a given model $f(\mathbf{x}; \boldsymbol{\theta})$, multiple empirical losses can be computed, one per objective as $\mathcal{L}_j(\boldsymbol{\theta}) = \mathbb{E}_{(\boldsymbol{x}, y^j) \in \mathcal{D}}[\mathcal{L}(f(\boldsymbol{x}; \boldsymbol{\theta}), y^j)]$. The model can then be optimized by minimizing a single loss, which is obtained by combining all the losses through a weighted summation as: $\min_{\boldsymbol{\theta}} \sum_{j=1}^{n} w^j \mathcal{L}_j(\boldsymbol{\theta})$.

A fundamental question is how to assign these weights and how to regulate the learning process to do well on all the objectives concurrently. Earlier research sought to address this issue based on assumptions regarding the efficacy of certain model architectures or optimization constraints, which may not be generally valid and can significantly increase complexity Zhang & Yang (2021).

In contrast, we propose to approach the learning and optimization for multi-objective learning from a different perspective. Specifically, we posit that the interaction process between the agent and the environment can be formalized as an Multi-Objective Markov Decision Process (MOMDP) Roijers et al. (2013). Denote the interaction trajectories collected by an existing agent as $\mathcal{D} = \{\tau_i\}_{i \in [1,M]}$. In the context of recommender systems, each trajectory $\tau$ records a complete recommendation session between a user entering and exiting the recommender system, such that $\tau = \{(\boldsymbol{s}_t, a_t, \boldsymbol{r}_t)\}_{t \in [1,|\tau|]}$. A state $\boldsymbol{s}_t \in \mathcal{S}$ is taken as the representation of user's preferences at a given timestep $t$. An action $a_t$ is recommended from the action space $\mathcal{A}$ which includes all candidate items, such that $|\mathcal{A}| = |\mathcal{V}| = N$ where $\mathcal{V}$ denotes the set of all items. $R(s_t, a_t)$ is the reward function, where $\boldsymbol{r}_t = R(s_t, a_t)$ means the agent receives a reward $\boldsymbol{r}_t$ after taking an action $a_t$ under state $\boldsymbol{s}_t$. Note that the reward function $R(\boldsymbol{s}_t, a_t)$ in MOMDP is represented by a multi-dimensional vector instead of a scalar.

In this context, all the objectives can be quantified using reward $\boldsymbol{r}_t$ at each time step. Specifically, since $\boldsymbol{r}_t$ is determined by user's behavior in response to recommended items, it naturally reflects the recommender's performance on these objectives. For example, if the user clicks the recommended item $a_t$, the value on the corresponding dimension of $\boldsymbol{r}_t$ can be set to 1; otherwise, it remains 0. In sequential recommendation scenarios, the target of the agent is to pursue better performance at the session level. Session-level performance can be evaluated by the cumulative reward from the current timestep to the end of the trajectory:

$$\boldsymbol{g}_t = \sum_{t'=t}^{|\tau|} \boldsymbol{r}_{t'}, \tag{1}$$

---

**Algorithm 1:** Training of MOGCSL

---

1  **Input:** training data $\mathcal{D}_{tr}$, batch size $B$, model parameters $\theta$
2  **Intialization:** initialize parameters $\theta$
3  Relabel all the rewards with goals according to Eq. (1)
4  **repeat**
5  $\quad$ Randomly sample a batch of $(\boldsymbol{s}_t, a_t, \boldsymbol{g}_t)$ from $\mathcal{D}_{tr}$
6  $\quad$ Compute the representation $\boldsymbol{M}_{\boldsymbol{s}_t, \boldsymbol{g}_t}$ via Eq. (2)-Eq.(3)
7  $\quad$ Derive the prediction logits via Eq. (4)
8  $\quad$ Calculate the loss function $\mathcal{L}(\theta)$ via Eq. (5)
9  $\quad$ Update $\theta$ by minimizing $\mathcal{L}(\theta)$ with stochastic gradient descent: $\theta \leftarrow \theta - \eta \frac{\partial \mathcal{L}(\theta)}{\partial \theta}$
10 **until** *convergence*

---

where $\boldsymbol{g}_t$ can be called as a "goal" in the literature of GCSL Yang et al. (2022).

Then, the target of mutli-objective learning for recommender systems can be formulated as *developing a policy that achieves satisfactory performance (i.e., the goals) across multiple objectives in recommendation sessions*. In this research, we address this problem within the framework of GCSL. During the training stage, the aim is to determine the optimal action to take from a given current state in order to achieve the specified goal. The agent, denoted as $\pi_\theta$, is trained by maximizing the likelihood of trajectories in the offline training dataset $\mathcal{D}_{tr}$ through an autoregressive approach, expressed as $\arg\max_\theta \mathbb{E}_{\mathcal{D}_{tr}}[log\pi_\theta(a|\boldsymbol{s}, \boldsymbol{g})]$. Notably, there are no predefined constraints or assumptions governing the learning process. During the inference stage, when an achievable and desirable goal is specified, the model is expected to select an action based on the goal and the current state, with the aim of inducing behaviours to achieve that goal.

### 3.2  MOGCSL TRAINING

The initial step of MOGCSL training is relabeling the training data by substituting the rewards with goals. Specifically, for each trajectory $\tau \in \mathcal{D}_{tr}$, we replace every tuple $(\boldsymbol{s}_t, a_t, \boldsymbol{r}_t)$ with $(\boldsymbol{s}_t, a_t, \boldsymbol{g}_t)$, where $\boldsymbol{g}_t$ is defined according to Eq. (1). Subsequently, we employ a sequential model Kang & McAuley (2018) based on Transformer-encoder (denoted as *T-enc*) to encode the users' sequential behaviors and obtain state representations. We chose a transformer-based encoder due to its widely demonstrated capability in sequential recommendation scenarios Kang & McAuley (2018). Specifically, let the interaction history of a user up to time $t$ be denoted as $v_{1:t-1} = \{v_1, ..., v_{t-1}\}$. We first map each item $v \in \mathcal{V}$ into the embedding space, resulting in the embedding representation of the history: $\boldsymbol{e}_{1:t-1} = [\boldsymbol{e}_1, ..., \boldsymbol{e}_{t-1}]$. Then we encode $\boldsymbol{e}_{1:t-1}$ by *T-enc*. Since the current timestep $t$ is also valuable for estimating user's sequential behavior, we incorporate it via a timestep embedding denoted as $emb_t$ through a straightforward embedding table lookup operation. Similarly, we derive the embedding of the goal $emb_{\boldsymbol{g}_t}$ through a simple fully connected (FC) layer with a subsequent normalization module. The final representation of state $\boldsymbol{s}_t$ is derived by concatenating the sequential encoding, timestep embedding and goal embedding together:

$$emb_{\boldsymbol{s}_t} = Concat(\textit{T-enc}(\boldsymbol{e}_{1:t-1}), emb_t, emb_{\boldsymbol{g}_t}). \tag{2}$$

To better capture the mutual information, we feed the state embedding into a self-attention block:

$$\boldsymbol{M}_{\boldsymbol{s}_t, \boldsymbol{g}_t} = Atten(emb_{\boldsymbol{g}_t})). \tag{3}$$

Then we use an MLP to map the derived embedding into the action space, where each logit represents the preference of taking a specific action (i.e., recommending an item):

$$[\pi_\theta(v^1|\boldsymbol{s}_t, \boldsymbol{g}_t), ..., \pi_\theta(v^N|\boldsymbol{s}_t, \boldsymbol{g}_t)] = \delta(MLP(\boldsymbol{M}_{\boldsymbol{s}_t, \boldsymbol{g}_t})), \tag{4}$$

where $v^i$ denotes the $i$-th item in the candidate pool, $\delta$ is the soft-max function, and $\theta$ denotes all parameters of this model. The model structure is shown in Appendix 1.

The training objective is to correctly predict the subsequent action that is mostly likely lead to a specific goal given the current state. As discussed in Section 3.1, each trajectory of user's interaction history represents a successful demonstration of reaching the goal that it actually achieved. As a result, the model can be naturally optimized by minimizing the expected cross-entropy as:

$$\mathcal{L}(\theta) = \mathbb{E}_{(\boldsymbol{s}_t, a_t, \boldsymbol{g}_t) \in \mathcal{D}_{tr}}[-log(\pi_\theta(a_t|\boldsymbol{s}_t, \boldsymbol{g}_t))]. \tag{5}$$

---

**Algorithm 2:** Inference of MOGCSL

---

1  **Input:** state $s'$, sample size $K$, policy model $\pi$, utility principle $U$, prior $q(g'|s')$, distribution of
    achievable goals $P(g^a|s', g', \pi)$

2  **Intialization:** set of potential input goals $G' = \emptyset$, set of expected achievable goals $G^a = \emptyset$

3  **for** $k = 1, \ldots, K$ **do**

4     Sample a $g'_k$ from $q(g'|s')$

5     Compute the expectation of the achievable goal through sampling: $\tilde{g}^a_k = \mathbb{E}_{g^a_k \sim P(\cdot|s', g'_k, \pi)} g^a_k$

6     $G' = G' \cup g'_k$

7     $G^a = G^a \cup \tilde{g}^a_k$

8  Choose the best $\tilde{g}^a_b$ from $G^a$ according to $U(\tilde{g}^a)$

9  Choose corresponding $g'_b$ from $G'$

10  **Return:** $\pi(\cdot|s', g'_b)$

---

The training process is illustrated in Algorithm 1.

### 3.3  MOGCSL INFERENCE

After training, we derive a model $\pi_\theta(a|s, g)$ that predicts the next action based on the given state and goal. However, while the goal can be accurately computed through each trajectory in the training data via Eq. (1), we must assign a desirable goal as input for the new state encountered during inference. GCSL approaches typically determine this goal-choosing strategy based on simple statistics calculated from the training data. E.g., Chen et al. (2021b) and Zheng et al. (2022) set the goals for all states at inference as the product of the maximal cumulative reward in training data and a fixed factor serving as a hyperparameter. Similarly, Xin et al. (2022) derive the goals for inference at a given timestep by scaling the mean of the cumulative reward in training data at the same timestep with a pre-defined factor. However, a central yet unexplored question is: *what are the general characteristics of the goals and how can we determine them for inference in a principled manner?*

In this paper, we investigate the distribution of the multi-dimensional goals that can be achieved during inference by first stating the following theorem. Proof is given in Appendix A.

**Theorem 1.** *Assume that the environment is modeled as an MOMDP. Consider a trajectory $\tau$ that is generated by the policy $\pi(a|s, g)$ given the initial state $s_1$ and goal $g_1$, the distribution of goals $g^a$ (i.e., cumulative rewards) that the agent actually achieves throughout the trajectory is determined by $(s_1, g_1, \pi)$.*

Based on this theorem, we'd like to learn the distribution of $g^a$ conditioned on $(s_1, g_1, \pi)$, denoted as $P(g^a|s_1, g_1, \pi)$. General generative models, such as GANs Mirza & Osindero (2014) and diffusion models Ho & Salimans (2022), can be employed to learn this distribution. In this paper, we propose the use of a conditional variational auto-encoder (CVAE) Sohn et al. (2015) due to its simplicity, robustness, and ease of formulation.

Specifically, this distribution can be learned directly on the training data $\mathcal{D}_{tr}$. For each $(s, g) \in \mathcal{D}_{tr}$, $g$ should be a sample from the distribution of the achievable goals by the policy $\pi$, given the initial state $s$ and input goal $g$. That's because the policy $\pi$ is trained to imitate the actions demonstrated by each data point in $\mathcal{D}_{tr}$, where the achieved goal of the trajectory starting from $(s, g)$ is exactly $g$. Let $c = (s, g, \pi)$. The loss function is:

$$\mathcal{L}_{CVAE1} = \mathbb{E}_{(s,g) \in \mathcal{D}_{tr}, z \sim Q_1}[log P_1(g|z, c) + D_{KL}(Q_1(z|g, c)||P(z))], \quad (6)$$

where $Q_1(z|g, c)$ is the encoder and $P_1(g|z, c)$ is the decoder. Based on Gaussian distribution assumption, they can be written as $Q_1 = \mathcal{N}(\mu(g, c), \Sigma(g, c))$ and $P_1 = \mathcal{N}(f_{CVAE1}(z, c), \sigma^2 I)$ respectively, where $z \sim \mathcal{N}(0, I)$. Then we can derive a sample of $g^a$ by inputting a sampled $z$ into $f_{CVAE1}$.

On the inference stage, given a new state $s'$, we first sample a set of goals $g'$ as the possible input of $\pi$ through a learnable prior $q(g'|s')$. Similarly, we learn this prior via another CVAE on the training data. The loss is:

$$\mathcal{L}_{CVAE2} = \mathbb{E}_{(s,g) \in \mathcal{D}_{tr}, z \sim Q_2}[log P_2(g|z, s) + D_{KL}(Q_2(z|g, s)||P(z))]. \quad (7)$$

Finally, we'll choose a desirable goal as the input along with the new state $s'$ encountered in inference by sampling from the two CVAE models. Specifically, we propose to: (1) sample from the prior $q(g'|s')$ to get a set of potential input goals, denoted as $G'$, (2) for each $g' \in G'$, estimate the expectation of the actually achievable goal $\tilde{g}^a$ by sampling from $P(\cdot|s', g', \pi)$ and taking the average, (3) choose a best goal as input for inference from $G'$ according to the associated expected $\tilde{g}^a$ by a predefined utility principle $U(\tilde{g}^a)$, which generally tends to pick up a "high" goal to achieve larger rewards on multiple objectives. The exact definition can be customized based on specific business requirements or objective priorities. E.g., choose by a predefined partial ordering [1]. Note that our algorithm is general and can accommodate adaptations of the utility function, as the desired goals are always set among the set of achievable goals, which naturally resolves the objective-conflicting issue. This definition flexibility allows practitioners to easily tailor MOGCSL to various practical scenarios and objective priorities. See Algorithm 2 for detailed inference pseudocode.

### 3.4 Analysis of Denoising Capability

An important benefit of MOGCSL is its capability to remove harmful effects of potentially noisy instances in the training data by leveraging the multiple-objective goals. To illustrate this, we consider the following setup that is common in recommender systems. There is a recommender system that has been operational, and recording data. At each interaction, the system shows the user a short list of items. The user then chooses one of these items. In the counterfactual that the recommender system is ideal, the action recorded would be $a$ which reveals the user's true interest. Since the actual recommender system to collect the data is not ideal, we have no direct access to $a$, but rather to a noisy version of it $\varepsilon(a)$.

We assume that the noisy portion of the training data originates from users who are presented with a list of items that are not suitable for them, rendering their reactions to these recommendations uninformative. Conversely, interactions achieving higher goals are generally less noisy, meaning $\varepsilon(a)$ is closer to $a$. To illustrate this, consider a scenario where a user clicks two recommended items ($v_1$ and $v_2$) under the same state. After clicking $v_1$, the user chooses to quit the system, while he stays longer and browses more items after clicking $v_2$. This indicates that the goal (i.e., cumulative reward) with $v_2$ is larger than that with $v_1$. In this case, we argue that $v_2$ should be considered as the user's truly preferred item over $v_1$. That's because the act of quitting, which results in a smaller goal, indicates user dissatisfaction with the previous recommendation, even though he did click $v_1$ before. Our proposed MOGCSL can model and leverage this mechanism based on multi-dimensional goals, which serve as a description of the **future effects of current actions on multiple objectives**. Specifically, by incorporating multi-dimensional goals as input, MOGCSL can effectively differentiate between noisy and noiseless samples in the training data. During inference, when high goals are specified as input, the model can make predictions based primarily on the patterns learned from the corresponding noiseless interaction data. It's worth noting that several prior studies on recommender system denoising Zhang et al. (2025); Chen et al. (2021a) have highlighted that such noises (i.e., users' noisy instant behaviors that don't align with their long-term, real interests) are very common and widely exist in real-world recommendation datasets.

To empirically validate this effect, we conduct experiments that are illustrated in Appendix B.2 due to limited space. The results demonstrate the denoising capability of MOGCSL.

## 4 Experiments

In this section, we introduce our experiments on two e-commerce datasets, aiming to address the following research questions: 1) **RQ1.** How does MOGCSL perform when compared to previous methods for multi-objective learning in recommender systems? 2) **RQ2.** How does MOGCSL mitigate the complexity challenges, including space and time complexity, as well as the intricacies of parameter tuning encountered in prior research? 3) **RQ3.** How does the goal-generation module for inference perform when compared to strategies based on simple statistics?

---

[1]See Section 4.4 for the implementation in our experiments.

### 4.1 EXPERIMENTAL SETUP

**Datasets** We use two public datasets: Challenge15 and RetailRocket. Both of them include binary labels indicating whether a user clicked or purchased the currently recommended item. More details of the datasets, metrics, and implementation specifics are provided in Appendix B.

**Baselines** Prior research on multi-objective learning encompass both model structure adaptation and optimization constraints. In our experiments, we consider two representative model architectures: Shared-Bottom Ma et al. (2018) and MMOE Ma et al. (2018). For works on optimization constraints, we compare four methods: Fixed-Weights Wang et al. (2016) assigns fixed weights for different objectives based on grid search; DWA Liu et al. (2019) dynamically adjusts weights by considering the dynamics of loss change; PE Lin et al. (2019) generates Pareto-efficient recommendations across multiple objectives; Nash-MTL Navon et al. (2022) characterize the Nash bargaining solution for multi-objective learning; FAMO Liu et al. (2024) adjusts weights to achieve balanced task loss reduction while maintaining relatively low space and time complexity.

Following previous researchYu et al. (2020), we consider all these optimization methods for each model architecture, resulting in eight baselines denoted as Share-Fix, Share-DWA, Share-PE, Share-Nash-MTL, Share-FAMO, MMOE-Fix, MMOE-DWA, MMOE-PE, MMOE-Nash-MTL, MMOE-FAMO. Additionally, we introduce a variant of a recent work called PRL Xin et al. (2022), which firstly applied GCSL to recommender systems. Specifically, similar to classic multi-objective methods, we compute the weighted summation of rewards from different objectives at each timestep. Then the overall cumulative reward is calculated as the goal, which is a scalar following conventional GCSL. We call this variant as MOPRL. Since we formulate our problem as an MOMDP, we also incorporate a recent baseline RMTL Liu et al. (2023), which applies offline reinforcement learning for sequential recommendation. Similarly, the aggregate reward is derived by the weighted summation of all objective rewards. Additionally, we compare against PMORS Jin et al. (2024), a recent supervised multi-objective learning framework designed for recommender systems. Note that to ensure a fair comparison, we employ the *T-enc* and self-attention block introduced in Section 3.2 as the base module to encode sequential data for all compared baselines.

**Evaluation Metrics** We employ two widely recognized information retrieval metrics to evaluate model performance in top-$k$ recommendation: Hit Ratio (HR@$k$) and Normalized discounted cumulative gain (NDCG@$k$). We use the abbreviation NG to denote NDCG in the tables. For each experiment, the mean and standard deviation over 5 seeds are reported.

### 4.2 PERFORMANCE COMPARISON (RQ1)

We begin by conducting experiments to compare the performance of MOGCSL and selected baselines in terms of top-$k$ recommendation. The experimental results are presented in Table 1. It's worth mentioning that a straightforward strategy based on training set statistics is employed to determine the inference goals in PRL Xin et al. (2022). Specifically, at each timestep in inference, the goal are set as the average cumulative reward from offline data at the same timestep, multiplied by a hyper-parameter factor $\lambda$ that is tuned using the validation set. To ensure a fair and meaningful comparison, we adopt the same strategy here for determining inference goals in MOGCSL. The comparison between different goal-choosing strategies is discussed in Section 4.4.

On RetailRocket, MOGCSL significantly outperforms previous multi-objective benchmarks in terms of purchase-related metrics. Regarding click metrics, MOGCSL achieves the best performance on HR, while Share-PE slightly outperforms it on NDCG. However, the performance gap between Share-PE and MOGCSL for purchase-related metrics ranges from 17% to 20%, whereas Share-PE only marginally outperforms MOGCSL on NDCG for purchase by less than 1%. Additionally, we observe that the more complex architecture design, MMOE, can perform worse than the simpler Shared-Bottom structure in many cases. Surprisingly, a naive optimization strategy based on fixed loss weights can outperform more advanced methods like DWA across several metrics (e.g., Share-Fix vs Share-DWA). These findings highlight the limitations of previous approaches that rely on assumptions about model architectures or optimization constraints, which may not be necessarily true in general environments. Similar trends are observed on Challenge15. While MMOE-PE performs slightly better on click metrics by 1-2%, MOGCSL achieves a substantial performance improvement on the more important purchase metrics by 11-20%. Especially, PE optimizes for Pareto efficiency by assuming that improving one objective should not degrade performance on any other objective.

Table 1: Comparison between MOGCSL and other baselines on RetailRocket and Challenge15 datasets. The mean and standard deviation over 5 seeds are reported. Boldface denotes the best results.

| [RetailRocket] | Purchase (%) | | | | Click (%) | | | |
|---|---|---|---|---|---|---|---|---|
| | HR@10 | NG@10 | HR@20 | NG@20 | HR@10 | NG@10 | HR@20 | NG@20 |
| Share-Fix | $48.57_{\pm0.17}$ | $45.79_{\pm0.09}$ | $49.47_{\pm0.10}$ | $46.01_{\pm0.08}$ | $35.51_{\pm0.24}$ | $25.85_{\pm0.16}$ | $40.15_{\pm0.20}$ | $27.03_{\pm0.14}$ |
| Share-DWA | $48.11_{\pm0.04}$ | $45.83_{\pm0.08}$ | $48.64_{\pm0.08}$ | $45.96_{\pm0.06}$ | $34.20_{\pm0.27}$ | $25.53_{\pm0.12}$ | $38.43_{\pm0.31}$ | $26.60_{\pm0.13}$ |
| Share-PE | $48.67_{\pm0.12}$ | $45.94_{\pm0.03}$ | $49.42_{\pm0.04}$ | $46.13_{\pm0.02}$ | $35.69_{\pm0.08}$ | $\mathbf{26.20_{\pm0.08}}$ | $40.28_{\pm0.11}$ | $\mathbf{27.37_{\pm0.07}}$ |
| Share-Nash | $48.84_{\pm0.21}$ | $46.23_{\pm0.17}$ | $50.07_{\pm0.19}$ | $46.50_{\pm0.22}$ | $35.87_{\pm0.20}$ | $26.04_{\pm0.16}$ | $40.52_{\pm0.25}$ | $27.11_{\pm0.13}$ |
| Share-FAMO | $48.92_{\pm0.17}$ | $46.11_{\pm0.10}$ | $50.19_{\pm0.09}$ | $46.78_{\pm0.13}$ | $35.97_{\pm0.14}$ | $26.17_{\pm0.09}$ | $40.71_{\pm0.11}$ | $27.35_{\pm0.18}$ |
| MMOE-Fix | $47.74_{\pm0.09}$ | $44.01_{\pm0.05}$ | $48.61_{\pm0.11}$ | $44.23_{\pm0.04}$ | $35.29_{\pm0.16}$ | $25.67_{\pm0.09}$ | $40.04_{\pm0.26}$ | $26.87_{\pm0.11}$ |
| MMOE-DWA | $47.78_{\pm0.40}$ | $44.57_{\pm0.13}$ | $48.44_{\pm0.24}$ | $44.79_{\pm0.09}$ | $35.68_{\pm0.46}$ | $26.13_{\pm0.29}$ | $40.22_{\pm0.57}$ | $27.28_{\pm0.32}$ |
| MMOE-PE | $46.58_{\pm0.22}$ | $43.72_{\pm0.15}$ | $47.37_{\pm0.11}$ | $43.94_{\pm0.14}$ | $35.39_{\pm0.27}$ | $26.19_{\pm0.11}$ | $39.78_{\pm0.39}$ | $27.31_{\pm0.14}$ |
| MMOE-Nash | $47.88_{\pm0.27}$ | $46.49_{\pm0.29}$ | $51.17_{\pm0.22}$ | $47.10_{\pm0.31}$ | $35.79_{\pm0.18}$ | $26.02_{\pm0.27}$ | $41.03_{\pm0.20}$ | $26.39_{\pm0.28}$ |
| MMOE-FAMO | $47.93_{\pm0.32}$ | $46.42_{\pm0.23}$ | $51.24_{\pm0.19}$ | $47.15_{\pm0.21}$ | $35.92_{\pm0.21}$ | $26.14_{\pm0.17}$ | $41.15_{\pm0.39}$ | $26.62_{\pm0.14}$ |
| RMTL | $62.84_{\pm0.24}$ | $49.92_{\pm0.20}$ | $67.04_{\pm0.15}$ | $50.89_{\pm0.25}$ | $33.95_{\pm0.35}$ | $24.23_{\pm0.41}$ | $39.87_{\pm0.32}$ | $25.27_{\pm0.29}$ |
| PMORS | $63.14_{\pm0.15}$ | $51.02_{\pm0.17}$ | $67.45_{\pm0.20}$ | $52.07_{\pm0.13}$ | $34.16_{\pm0.26}$ | $24.09_{\pm0.23}$ | $39.86_{\pm0.19}$ | $25.31_{\pm0.27}$ |
| MOPRL | $61.18_{\pm0.19}$ | $50.74_{\pm0.10}$ | $64.76_{\pm0.25}$ | $51.65_{\pm0.02}$ | $33.99_{\pm0.11}$ | $24.31_{\pm0.08}$ | $38.98_{\pm0.08}$ | $25.57_{\pm0.08}$ |
| MOGCSL | $\mathbf{65.43_{\pm0.15}}$ | $\mathbf{52.92_{\pm0.11}}$ | $\mathbf{69.28_{\pm0.14}}$ | $\mathbf{53.90_{\pm0.14}}$ | $\mathbf{36.30_{\pm0.25}}$ | $25.24_{\pm0.15}$ | $\mathbf{41.92_{\pm0.55}}$ | $26.67_{\pm0.19}$ |

| [Challenge15] | Purchase (%) | | | | Click (%) | | | |
|---|---|---|---|---|---|---|---|---|
| | HR@10 | NG@10 | HR@20 | NG@20 | HR@10 | NG@10 | HR@20 | NG@20 |
| Share-Fix | $38.18_{\pm0.10}$ | $25.47_{\pm0.26}$ | $43.97_{\pm0.18}$ | $26.93_{\pm0.20}$ | $41.61_{\pm0.30}$ | $25.77_{\pm0.16}$ | $49.19_{\pm0.43}$ | $27.70_{\pm0.19}$ |
| Share-DWA | $38.27_{\pm0.18}$ | $25.63_{\pm0.08}$ | $43.95_{\pm0.19}$ | $27.07_{\pm0.08}$ | $41.49_{\pm0.24}$ | $25.90_{\pm0.16}$ | $48.90_{\pm0.20}$ | $27.77_{\pm0.14}$ |
| Share-PE | $38.92_{\pm0.09}$ | $25.83_{\pm0.13}$ | $44.82_{\pm0.12}$ | $27.32_{\pm0.07}$ | $42.46_{\pm0.16}$ | $26.39_{\pm0.06}$ | $50.05_{\pm0.17}$ | $28.32_{\pm0.06}$ |
| Share-Nash | $39.12_{\pm0.17}$ | $25.87_{\pm0.15}$ | $45.13_{\pm0.22}$ | $27.56_{\pm0.14}$ | $43.10_{\pm0.24}$ | $26.89_{\pm0.19}$ | $50.61_{\pm0.17}$ | $28.88_{\pm0.21}$ |
| Share-FAMO | $39.06_{\pm0.13}$ | $25.94_{\pm0.21}$ | $44.97_{\pm0.18}$ | $27.65_{\pm0.09}$ | $43.25_{\pm0.13}$ | $27.02_{\pm0.11}$ | $50.58_{\pm0.26}$ | $28.94_{\pm0.14}$ |
| MMOE-Fix | $35.34_{\pm0.12}$ | $23.87_{\pm0.07}$ | $40.68_{\pm0.09}$ | $25.22_{\pm0.12}$ | $43.82_{\pm0.16}$ | $27.33_{\pm0.09}$ | $51.42_{\pm0.19}$ | $29.26_{\pm0.10}$ |
| MMOE-DWA | $37.04_{\pm0.40}$ | $24.88_{\pm0.13}$ | $42.64_{\pm0.24}$ | $26.30_{\pm0.09}$ | $42.20_{\pm0.46}$ | $26.45_{\pm0.29}$ | $49.48_{\pm0.57}$ | $28.30_{\pm0.32}$ |
| MMOE-PE | $36.40_{\pm0.36}$ | $24.66_{\pm0.19}$ | $41.52_{\pm0.33}$ | $25.96_{\pm0.19}$ | $\mathbf{44.04_{\pm0.09}}$ | $\mathbf{27.44_{\pm0.03}}$ | $\mathbf{51.60_{\pm0.07}}$ | $\mathbf{29.37_{\pm0.03}}$ |
| MMOE-Nash | $38.01_{\pm0.41}$ | $25.40_{\pm0.32}$ | $43.69_{\pm0.27}$ | $27.06_{\pm0.29}$ | $43.77_{\pm0.41}$ | $26.87_{\pm0.52}$ | $50.96_{\pm0.59}$ | $28.92_{\pm0.50}$ |
| MMOE-FAMO | $37.92_{\pm0.56}$ | $25.43_{\pm0.22}$ | $43.63_{\pm0.29}$ | $27.12_{\pm0.15}$ | $43.71_{\pm0.55}$ | $26.98_{\pm0.38}$ | $50.98_{\pm0.66}$ | $29.03_{\pm0.39}$ |
| RMTL | $53.34_{\pm0.46}$ | $33.65_{\pm0.36}$ | $62.89_{\pm0.49}$ | $34.74_{\pm0.37}$ | $41.41_{\pm0.41}$ | $24.87_{\pm0.38}$ | $49.29_{\pm0.43}$ | $26.91_{\pm0.29}$ |
| PMORS | $54.98_{\pm0.31}$ | $34.77_{\pm0.28}$ | $63.52_{\pm0.24}$ | $37.05_{\pm0.35}$ | $42.36_{\pm0.39}$ | $25.10_{\pm0.42}$ | $50.07_{\pm0.30}$ | $26.84_{\pm0.26}$ |
| MOPRL | $54.79_{\pm0.37}$ | $35.37_{\pm0.26}$ | $63.10_{\pm0.45}$ | $37.49_{\pm0.27}$ | $42.14_{\pm0.21}$ | $25.62_{\pm0.18}$ | $50.18_{\pm0.25}$ | $27.66_{\pm0.19}$ |
| MOGCSL | $\mathbf{56.82_{\pm0.25}}$ | $\mathbf{35.93_{\pm0.15}}$ | $\mathbf{65.64_{\pm0.55}}$ | $\mathbf{38.17_{\pm0.19}}$ | $42.47_{\pm0.15}$ | $25.64_{\pm0.11}$ | $50.52_{\pm0.14}$ | $27.73_{\pm0.11}$ |

However, this assumption does not align well with the realities of recommender systems. In practice, a significant improvement in high-value objectives (such as purchase) accompanied by only minor reductions in less critical objectives (such as clicks) is often preferable in practical scenarios.

Apart from previous benchmarks for multi-objective learning, MOGCSL also exhibits significant and consistent performance improvements on both datasets compared to offline RL based RMTL and the variant MOPRL created on standard GCSL. Especially, at each timestep, the overall reward of them is calculated as the weighted sum of rewards across all objectives. In our experiments, it's defined as $r' = w_c r^c + w_p r^p$, where $r^c$ and $r^p$ are click and purchase reward and $w_c + w_p = 1$. Then the goal in MOPRL is derived by calculating the cumulative rewards as a scalar. In contrast, MOGCSL takes the goal as a vector, allowing the disentanglement of rewards for different objectives along different dimensions. Notably, no additional summation weights or other constraints are required. We have conducted experiments to compare the performance of MOGCSL to MOPRLs with different weight combinations. The results show that MOGCSL consistently outperforms MOPRL across all weight combinations on both click and purchase metrics, demonstrating that representing the goal as a multi-dimensional vector enhances the effectiveness of GCSL on multi-objective learning. See the figure for the comparison in Appendix B.5.

## 4.3 COMPLEXITY COMPARISON (RQ2)

Apart from the performance improvement, MOGCSL also benefits from seamless integration with classic sequential models, adding minimal additional complexity. During the training

stage, the only extra complexity arises from relabeling one-step rewards with goals and including them as input to the sequential model. In contrast, previous multi-objective learning methods often introduce significantly excess time and space complexity Zhang & Yang (2021). For instance, MMOE and Shared-Bottom both design separate towers for each objective Ma et al. (2018), leading to a significant increase in model parameters as the number of tasks grows. RMTL needs an additional RL head for optimization on TD error. MOGCSL, in the other hand, only requires a simple MLP layer for action projection. In terms of time complexity, DWA, Nash-MTL and FAMO require recording and calculating loss change dynamics for each training epoch, while PE and PMORS involve computing the inverse of a large parameter matrix to solve an optimization problem under KKT conditions. Additionally, tuning the weight combinations for multiple objectives using grid search in methods like Fix-Weight and MO-PRL is highly time-consuming, requiring approximately $O(m^n)$ repetitive experiments to identify a near-optimal combination, where $m$ is the size of the search space per dimension and $n$ is the number of objectives. In contrast, MOGCSL inherently avoids this weight-tuning process.

Table 2: Comparison of time and space complexity on RetailRocket.

|  | Model Size | Training time |
| --- | --- | --- |
| Share-Fix | 14.0M | 9.6Ks |
| Share-DWA | 14.0M | 5.3Ks |
| Share-PE | 14.0M | 5.6Ks |
| Share-Nash-MTL | 14.0M | 8.6Ks |
| Share-FAMO | 14.0M | 5.1Ks |
| MMOE-Fix | 14.1M | 10.2Ks |
| MMOE-DWA | 14.1M | 9.5Ks |
| MMOE-PE | 14.1M | 60.5Ks |
| MMOE-Nash-MTL | 14.1M | 12.4Ks |
| MMOE-FAMO | 14.1M | 8.8Ks |
| RMTL | 17.5M | 100.2Ks |
| PMORS | 14.2M | 10.4Ks |
| MOPRL | 9.1M | 3.2Ks |
| MOGCSL | 9.1M | 3.0Ks |

Table 2 summarizes the complexity comparison. It's evident that MOGCSL significantly benefits from a smaller model size and faster training speed, while concurrently achieving great performance.

### 4.4 GOAL-GENERATION STRATEGY COMPARISON (RQ3)

As introduced in Section 3.3, most previous research on GCSL decides the inference goals based on simple statistics on the training set. However, we demonstrate that the distribution of the goals achieved by the agent during inference should be jointly determined by the initial state, input goal and behavior policy. Based on that, we propose a novel algorithm (see Algorithm 2) that leverages CVAE to derive feasible and desirable goals for inference. Note that an utility principle $U(\boldsymbol{g})$ is required to evaluate the goodness of the multi-dimensional goals, which is generally preferable for "high" goals but could be flexible with specific business requirements. In our experiments, we select the best goal $\tilde{\boldsymbol{g}}_b^a$ from the set $\boldsymbol{G}^a$ based on the following rule, which ensures that no goal within the achievable set is superior to the selected goal across all objectives:

$$\tilde{\boldsymbol{g}}_b^a = \tilde{\boldsymbol{g}} \in \boldsymbol{G}^a, \text{ s.t. } \nexists \tilde{\boldsymbol{g}}' \in \boldsymbol{G}^a \setminus \tilde{\boldsymbol{g}}, \ \tilde{g}_i' \geq \tilde{g}_i \ \forall i \in [1, d]. \tag{8}$$

We compare two variants of MOGCSL here. MOGCSL-S employs the statistical strategy introduced in Section 4.2, while MOGCSL-C utilizes the goal-choosing method based on CVAE (Algorithm 2). As shown in Table 3, we observe that these two strategies do not significantly differ in overall performance across both datasets. While MOGCSL-C performs slightly better on RetailRocket, it exhibits worse performance on Challenge15. To investigate the reason, we conduct an additional experiment by varying the factor $\lambda$ for the inference goals of MOGCSL-S. The results reveals that the optimal performance is achieved when the factor is set between 1 and 2 for all metrics. When it grows larger, performance consistently declines. The figure is shown in Appendix B.6. Interestingly, similar findings have been reported in prior research Chen et al. (2021b); Xin et al. (2022); Zheng et al. (2022), demonstrating that setting very large inference goals can indeed harm performance.

We posit that the sparsity of training data within the high-goal space may contribute to the suboptimal performance of more advanced goal-choosing methods. While we may find some potentially achievable high goals, the model lacks sufficient training data to learn effective actions to reach these goals. Notably, the mean cumulative reward across all trajectories in both datasets is only around 5.3 for click and 0.2 for purchase. Consequently, most training data demonstrates how to achieve relatively low goals, hindering the model's ability to generalize effectively for larger goals in inference. To validate this, we conduct additional experiments on a dataset with higher average goals. Within our MOGCSL framework, we define click and like as the two dimensions of the multi-objective goals.

Table 3: Comparison between statistical strategy and CVAE-based method for goal-choosing.

| [RetailRocket] | Purchase (%) | | | | Click (%) | | | |
|---|---|---|---|---|---|---|---|---|
| | HR@10 | NG@10 | HR@20 | NG@20 | HR@10 | NG@10 | HR@20 | NG@20 |
| MOGCSL-S | **65.43**$_{\pm 0.15}$ | **52.92**$_{\pm 0.11}$ | 69.28$_{\pm 0.14}$ | 53.90$_{\pm 0.14}$ | 36.30$_{\pm 0.25}$ | 25.24$_{\pm 0.15}$ | 41.92$_{\pm 0.55}$ | 26.67$_{\pm 0.19}$ |
| MOGCSL-C | 65.01$_{\pm 0.07}$ | 52.89$_{\pm 0.04}$ | **69.34**$_{\pm 0.05}$ | **54.00**$_{\pm 0.04}$ | **36.54**$_{\pm 0.02}$ | **25.41**$_{\pm 0.04}$ | **42.20**$_{\pm 0.03}$ | **26.84**$_{\pm 0.06}$ |

| [Challenge15] | Purchase (%) | | | | Click (%) | | | |
|---|---|---|---|---|---|---|---|---|
| | HR@10 | NG@10 | HR@20 | NG@20 | HR@10 | NG@10 | HR@20 | NG@20 |
| MOGCSL-S | **56.82**$_{\pm 0.25}$ | **35.93**$_{\pm 0.15}$ | **65.64**$_{\pm 0.55}$ | **38.17**$_{\pm 0.19}$ | **42.27**$_{\pm 0.15}$ | **25.64**$_{\pm 0.11}$ | **50.52**$_{\pm 0.14}$ | **27.73**$_{\pm 0.11}$ |
| MOGCSL-C | 55.13$_{\pm 0.07}$ | 35.04$_{\pm 0.02}$ | 63.98$_{\pm 0.04}$ | 37.30$_{\pm 0.03}$ | 42.14$_{\pm 0.04}$ | 25.37$_{\pm 0.07}$ | 50.12$_{\pm 0.09}$ | 27.53$_{\pm 0.05}$ |

| [Tenrec] | Like (%) | | | | Click (%) | | | |
|---|---|---|---|---|---|---|---|---|
| | HR@10 | NG@10 | HR@20 | NG@20 | HR@10 | NG@10 | HR@20 | NG@20 |
| MOGCSL-S | 5.96$_{\pm 0.17}$ | 2.15$_{\pm 0.12}$ | 6.93$_{\pm 0.20}$ | 2.70$_{\pm 0.11}$ | 4.87$_{\pm 0.13}$ | 1.52$_{\pm 0.08}$ | 5.67$_{\pm 0.14}$ | 1.95$_{\pm 0.11}$ |
| MOGCSL-C | **6.78**$_{\pm 0.07}$ | **2.99**$_{\pm 0.02}$ | **7.84**$_{\pm 0.05}$ | **3.86**$_{\pm 0.04}$ | **5.66**$_{\pm 0.05}$ | **2.14**$_{\pm 0.05}$ | **6.73**$_{\pm 0.03}$ | **2.68**$_{\pm 0.07}$ |

The mean cumulative rewards across all trajectories are approximately 28.3 for click and 1.2 for like, both notably higher than those observed in the RetailRocket and Challenge15 datasets.

Table 3 shows the comparison results, where MOGCSL-C significantly outperforms MOGCSL-S. Together with previous experiments on datasets featuring lower average goals, these results reinforce our conclusion and provide clear practical insights into selecting between simple and advanced goal-selection strategies based on dataset properties. It's worth noting that a very recent work, RVDT Bai et al. (2025), provides an initial attempt at offering a more formal and theoretically grounded analysis of this problem. Leveraging insights and relevant theory from offline reinforcement learning, the authors draw similar conclusions and demonstrate that insufficient high-quality trajectories in offline datasets can lead to suboptimal outcomes when GCSL policies are overly optimistic.

Applying these insights to our setting, although the CVAE-based algorithm can identify potentially achievable high-value goals, the GCSL model may lack sufficient training data to learn effective actions for reaching these goals. Consequently, a more balanced strategy—such as selecting goals based on the average cumulative goals in the offline data, scaled by a hyperparameter factor—often proves more effective when high-goal training data is limited.

The results provide several insights for selecting goal-choosing strategies when applying MOGCSL in practical applications. First, strategies based on simple statistics on the training data prove to be efficient and effective in many cases, particularly when low latency or reduced model complexity is required during inference. Second, if we aim to further enhance performance using more advanced goal-choosing algorithms, access to a training set with more instances with high-valued goals could be crucial. Last, it's worth noting that our core technical contribution lies in developing a novel multi-objective learning framework leveraging GCSL, addressing and modeling the achievability of goals at inference. Within this framework, we explore and validate both principled (MOGCSL-C) and heuristic (MOGCSL-S) goal-selection methodologies to derive these achievable and "high" goals on multiple objectives. These methods demonstrate complementary strengths across diverse datasets and empirical conditions, which is itself a valuable and non-trivial finding.

## 5 CONCLUSION

In this work, we propose a novel framework named MOGCSL for multi-objective recommendation. MOGCSL utilizes a vectorized goal to disentangle the representation of different objectives. Building upon GCSL, it can be directly combined with conventional sequential models and optimized through supervised learning, without requiring handcrafted model architecture or optimization constraints. Beyond training process, we theoretically analyze the properties of inference goals and propose a novel goal-generation algorithm accordingly. Extensive experiments demonstrate the superiority of MOGCSL in both effectiveness and efficiency. For future work, we aim to explore a more effective goal-generation strategy for inference, which may necessitate a change in the training paradigm.

## REPRODUCIBILITY STATEMENT

We provide anonymized code at https://anonymous.4open.science/r/MOGCSL-D7A2 , together with the datasets used in our experiments. Comprehensive details of the experimental setup, including hyperparameters, training details, and evaluation methods, are presented in the Experiments section and Appendix. With these resources, we are confident that readers will be able to reproduce the results presented in the paper.

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

## A    PROOF OF THEOREM 1

We begin by first proving the following lemma.

**Lemma 1.** *Assume that the environment is modeled as am MOMDP. Consider a trajectory $\tau$ that is generated by the policy $\pi(a|s, g)$ given the initial state $s_1$ and goal $g_1$, the distributions of $s_t$ and $(s_t, g_t)$ at each timestep are both determined by $(s_1, g_1, \pi(a|s, g))$.*

*Proof.* First, for $t = 1$ we have:

$$\Pr(r_1 = r) = \sum_a \Pr(r|s_1, a) = r)\pi(a|s_1, g_1). \tag{9}$$

Note that the reward function $R(s, a)$ is fixed for the given environment. Then, we complete the proof by mathematical induction.

**Statement**:The distributions of $s_t$ and $(s_t, g_t)$ are both determined by $(s_1, g_1, \pi(a|s, g))$, for $t = 2, 3, ..., |\tau|$.

**Base case** $t = 2$: Since $s_1$ and $g_1$ are given and fixed, we have:

$$\Pr(s_2 = s) = \sum_a T(s|a, s_1)\pi(a|s_1, g_1). \tag{10}$$

It's clear that $s_2 \sim f_{s_2}(s; s_1, g_1, \pi)$ where $f_{s_2}$ is a distribution determined by $(s_1, g_1, \pi)$.

For $(s_2, g_2)$, according the definition of $g_t$ in Eq. (1), when a reward $r_t$ is received, the desired goal on next timestep is $g_{t+1} = g_t - r_t$. Combined with Eq. (9), We have:

$$\Pr(s_2 = s, g_2 = g) = \sum_a \Pr(g_1 - g|s_1, a) = g_1 - g)T(s|s_1, a)\pi(a|s_1, g_1). \tag{11}$$

Since the dynamic function $T(s'|s, a)$ is given, it's clear that $(s_2, g_2) \sim f_{s_2, g_2}(s, g; s_1, g_1, \pi)$.

**Inductive Hypothesis**: Suppose the statement holds for all $t$ up to some $n$, $2 \le n \le |\tau| - 1$.

**Inductive Step**: Let $t = n + 1$, similar to the base case, we have:

$$\Pr(s_{n+1} = s) = \sum_{a', s', g'} T(s|s', a')\pi(a'|s', g')\Pr(s_n = s', g_n = g'). \tag{12}$$

$$\Pr(s_{n+1} = s, g_{n+1} = g) = \sum_{a', s', g'} [\Pr(r' = g' - g|s', a')T(s|s', a') \\ \cdot \pi(a'|s', g')\Pr(s_n = s', g_n = g')]. \tag{13}$$

According to the hypothesis that the distributions of $(s_n, g_n)$ is determined by $(s_1, g_1, \pi(a|s, g))$, it's easy to see that $s_{n+1} \sim f_{g_{n+1}}(g; s_1, g_1, \pi)$ and $(s_{n+1}, g_{n+1}) \sim f_{s_{n+1}, g_{n+1}}(s, g; s_1, g_1, \pi)$.

As a result, the statement holds for $t = n + 1$. By the principle of mathematical induction, the statement holds for all $t = 2, 3, ..., |\tau|$. Apparently, that proves Lemma 1.    □

Then, based on the lemma, we can prove Theorem 1.

*Proof.* Let $|\tau| = T$, by definition we have:

$$g^a = \sum_{t=1}^{T} r_t, \tag{14}$$

Let $x_n = (r_n, ..., r_1)$, according to the Markov property and Bayes' rules we have:

$$P(r_n|r_{n-1}, ..., r_1) = P(r_n|x_{n-1})$$
$$= \sum_{s_n} P(r_n|s_n, x_{n-1})P(s_n|x_{n-1})$$
$$= \sum_{s_n} P(r_n|s_n) \sum_{s_{n-1}} P(s_n|s_{n-1}, x_{n-1})P(s_{n-1}|x_{n-1}) \tag{15}$$
$$= \sum_{s_n} P(r_n|s_n) \sum_{s_{n-1}} P(s_n|s_{n-1}, x_{n-1})... \sum_{s_2} P(s_3|s_2, x_{n-1})P(s_2|x_{n-1})$$

For the first term, we have:

$$P(\boldsymbol{r}_n|\boldsymbol{s}_n) = \sum_{a_n, \boldsymbol{g}_n} \frac{\Pr(\boldsymbol{r}_n = \boldsymbol{s}_n, a_n)\pi(a_n|\boldsymbol{s}_n, \boldsymbol{g}_n)P(\boldsymbol{s}_n, \boldsymbol{g}_n)}{P(\boldsymbol{s}_n)}. \tag{16}$$

Since $(\boldsymbol{s}_1, \boldsymbol{g}_1)$ is given and fixed, for each $m \in [2, n-1]$ we have:

$$P(\boldsymbol{s}_{m+1}|\boldsymbol{s}_m, \boldsymbol{x}_{n-1}) = \sum_{a_m} \pi(a_m|\boldsymbol{s}_m, \boldsymbol{g}_1 - \sum_{i=1}^{m-1} \boldsymbol{r}_i)T(\boldsymbol{s}_{m+1}|\boldsymbol{s}_m, a_m)\Pr(\boldsymbol{r}_m|\boldsymbol{s}_m, a_m). \tag{17}$$

Similarly, the last term can be written as:

$$P(\boldsymbol{s}_2|\boldsymbol{x}_{n-1}) = \sum_{a_1} \pi(a_1|\boldsymbol{s}_1, \boldsymbol{g}_1)T(\boldsymbol{s}_2|\boldsymbol{s}_1, a_1)\Pr(\boldsymbol{r}_1|\boldsymbol{s}_1, a_1). \tag{18}$$

According to Lemma 1, the distributions of $\boldsymbol{s}_n$ and $(\boldsymbol{s}_n, \boldsymbol{g}_n)$ are both determined by $(\boldsymbol{s}_1, \boldsymbol{g}_1, \pi(a|\boldsymbol{s}, \boldsymbol{g}))$. As a result, by Eq. (15 - 18), it's clear that the distribution of the conditional probability distribution $P(\boldsymbol{r}_{n+1}|\boldsymbol{r}_n, ..., \boldsymbol{r}_1)$ is also determined by $(\boldsymbol{s}_1, \boldsymbol{g}_1, \pi(a|\boldsymbol{s}, \boldsymbol{g}))$. Then, the distribution of $\boldsymbol{g}^a$ can be written as:

$$\begin{aligned}
\Pr(\boldsymbol{g}^a = \boldsymbol{g}) &= \int \cdots \int_{\sum_{i=1}^{T} \boldsymbol{r}_i = \boldsymbol{g}} f(\boldsymbol{r}_1, \boldsymbol{r}_2, ..., \boldsymbol{r}_T)d\boldsymbol{r}_1 d\boldsymbol{r}_2 ... d\boldsymbol{r}_T \\
&= \int \cdots \int_{\sum_{i=1}^{T} \boldsymbol{r}_i = \boldsymbol{g}} f_1(\boldsymbol{r}_1)f_2(\boldsymbol{r}_2|\boldsymbol{r}_1)...f_T(\boldsymbol{r}_T|\boldsymbol{r}_{T-1}, ..., \boldsymbol{r}_1)d\boldsymbol{r}_1 ... d\boldsymbol{r}_T
\end{aligned} \tag{19}$$

Obviously, the distribution of $\boldsymbol{g}^a$ is determined by $(\boldsymbol{s}_1, \boldsymbol{g}_1, \pi(a|\boldsymbol{s}, \boldsymbol{g}))$, which is exactly what Theorem 1 states.

$\square$

# B  EXPERIMENT DETAILS

## B.1  MODEL STRUCTURE

The model structure of MOGCSL is shown in Figure 1.

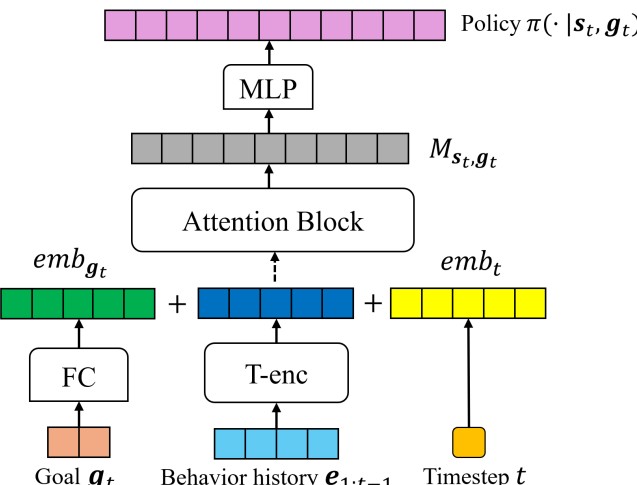

Figure 1: Model structure of MOGCSL.

Table 4: Comparison of XGBoost with different inputs. The mean and standard deviation over 5 seeds are reported.

|  | Accuracy | M-Logloss |
| --- | --- | --- |
| XGBoost-s | 0.0576$\pm$0.0038 | 3.7595$\pm$0.0393 |
| XGBoost-ug | 0.0598$\pm$0.0009 | 3.7262$\pm$0.0281 |
| XGBoost-mg | **0.0634**$\pm$**0.0027** | **3.6603**$\pm$**0.0097** |

## B.2 DENOISING CAPABILITY EXPERIMENTS

To illustrate the denoising capability of MOGCSL, we consider the same set-up introduced in Section 3.4, including definitions and notation of state, action, and reward. As described before, we assume that the noisy portion of the training data originates from users who are presented with a list of items that are not suitable for them, rendering their choices for these recommendations not meaningful. Conversely, data samples with higher goals are generally less noisy, meaning $\varepsilon(a)$ is closer to $a$. We also record a long-term and multidimensional goal (e.g., the cumulative reward) $\boldsymbol{g} = (g_1, ..., g_n)$ at each interaction (known only at the end of the session, but recorded retroactively). Thus our training data is a sample from a distribution $\mathcal{D}$ of tuples $(\boldsymbol{s}, \boldsymbol{g}, \varepsilon(a))$, where the state of the user is represented by a vector $\boldsymbol{s}$.

To empirically show the effect of this phenomenon in a simple set-up, we generate a dataset as follows [2]: (1): The states and the goals are sampled from two independent multivariate normal distributions with dimension of 50 and 5 respectively. (2) The ground-truth action $a$ is entirely determined by $\boldsymbol{s}$, whose ID is set to the number of coordinates of $\boldsymbol{s}$ that are greater than 0. (3) Define $\varepsilon(a)$ as: $\varepsilon(a) = a$ if $g_i > -1$ for all $i$; otherwise $\varepsilon(a)$ is uniformly random.

Formally, $\varepsilon(a)$ is determined by $\boldsymbol{g}$ and $\boldsymbol{s}$ as follows:

$$\varepsilon(a) = (\prod_{i=0}^{n} \mathbb{1}(g_i > -1))a + (1 - \prod_{i=0}^{n} \mathbb{1}(g_i > -1))randint[1, N], \tag{20}$$

where $a = \sum_{j=0}^{l} \mathbb{1}(s_j > 0)$.

Since MOGCSL is applicable to any supervised model by integrating goals as additional input features, we choose a simple XGBoost classifier Chen & Guestrin (2016) here for the sake of clarity. Specifically, we train three variants of XGBoost classifier on this data to predict the action given a state: (1) XGBoost-s: this variant only takes $\boldsymbol{s}$ as input and ignores $\boldsymbol{g}$. It cannot detect noisy instances in $\mathcal{D}$ because it lacks access to $\boldsymbol{g}$. (2) XGBoost-ug: this variant is taken as a single-objective GCSL model, which takes $\boldsymbol{s}$ and only the first coordinate $g_1$ of $\boldsymbol{g}$ as input. Clearly, it's also unable to precisely distinguish noisy data since $\varepsilon(a)$ is determined by all dimensions of $\boldsymbol{g}$ (as shown in Eq. (20)). (3) XGBoost-mg: this variant is based on our multi-objective GCSL, which takes both $\boldsymbol{s}$ and $\boldsymbol{g}$ as input. It is the only one capable of distinguishing all the noisy data by learning the determination pattern from $\boldsymbol{g}$ and $\boldsymbol{s}$ to $\varepsilon(a)$.

During inference stage, for XGBoost-ug and XGBoost-mg, we adopt a simple strategy to determine the goals: directly setting each dimension of $\boldsymbol{g}$ to 1, which serves as a high value to satisfy the condition $g_i > -1$ for the noiseless samples where $\varepsilon(a) = a$.

The results are presented in Table 4. It is evident that XGBoost-mg achieves the best performance. By incorporating multi-dimensional goals as input, XGBoost-mg can effectively differentiate between noisy and noiseless samples in the training data based on MOGCSL. During inference, when a high goal is specified as input, the model can make predictions based solely on the patterns and knowledge learned from the noiseless data.

---

[2]We generate synthetic data due to the unavailability of labels distinguishing between random and genuinely preference-driven user interactions in real-world datasets.

### B.2.1 DATASETS

We conduct experiments on two publicly available datasets: Challenge15 [3] and RetailRocket [4]. They are both collected from online e-business platforms by recording users' sequential behaviours in recommendation sessions. Specifically, both datasets include binary labels indicating whether a user clicked or purchased the currently recommended item. Following previous research Xin et al. (2022; 2020), we filter out sessions with lengths shorter than 3 or longer than 50 to ensure data quality.

After preprocessing, the Challenge15 dataset comprises 200,000 sessions, encompassing 26,702 unique items, 1,110,965 clicks and 43,946 purchases. Similarly, the processed RetailRocket dataset consists of 195,523 sessions, involving 70,852 distinct items. It documents 1,176,680 clicks and 57,269 purchases. We partition them into training, validation, and test sets, maintaining an 8:1:1 ratio.

### B.2.2 BASELINE DETAILS

In the experiments, we compare two representative model architectures for multi-objective learning:

- **Shared-Bottom** Ma et al. (2018): A classic model structure for multi-objective learning. The bottom of the model is a neural network shared across all objectives. On top of this shared base, separate towers are added for each objective, producing predictions specific to that objective.

- **MMOE** Ma et al. (2018): A widely used multi-objective model architecture. It first maps inputs to multiple expert modules shared by all objectives. These experts contribute to each objective through designed gates. The final input for each tower is a weighted summation of the experts' outputs.

Beyond architectural adaptations, other works focus on studying optimization constraints, mainly through adjusting weights of losses for different objectives. We compare the following methods:

- **Fixed-Weights** Wang et al. (2016): A straightforward strategy that assigns fixed weights based on grid search results from the validation set. These weights remain constant throughout the whole training stage.

- **DWA** Liu et al. (2019): This method aims to dynamically assign weights by considering the rate of loss change for each objective during recent training epochs. Generally, it tends to assign larger weights to objectives with slower loss changes.

- **PE** Lin et al. (2019): It's designed for generating Pareto-efficient recommendations across multiple objectives. The model optimizes for Pareto efficiency, ensuring no further improvement in one objective comes at the expense of any others.

- **Nash-MTL** Navon et al. (2022): This method frames the gradient combination step in mulit-objective learning as a bargaining game and use the Nash bargaining solution to find the optimal update direction.

- **FAMO** Liu et al. (2024): This recent method aims to dynamically adjust the weights for multi-objective learning, achieving balanced task loss reduction while maintaining relatively low space and time complexity.

Note that to ensure a fair comparison, we employ the *T-enc* and self-attention block introduced in Section 3.2 as the base module to encode sequential data for all compared baselines.

### B.3 EVALUATION METRICS

We employ two widely recognized information retrieval metrics to evaluate model performance in top-$k$ recommendation. Hit Ratio (HR@$k$) is to quantify the proportion of recommendations where the ground-truth item appears in the top-$k$ positions of the recommendation list Hidasi et al. (2015). Normalized discounted cumulative gain (NDCG@$k$) further considers the positional relevance of

---

[3]https://recsys.acm.org/recsys15/challenge
[4]https://www.kaggle.com/retailrocket/ecommerce-dataset

ranked items, assigning greater importance to top positions during calculation Kang & McAuley (2018). Given our dual objectives in experiments, we evaluate performance using HR@$k$ and NDCG@$k$ based on corresponding labels for click and purchase events (i.e., whether an item was clicked or purchased by the user).

### B.4 IMPLEMENTATION DETAILS

First, to ensure a fair comparison, we employ the transformer encoder and self-attention block introduced in Section 3.2 as the base module to encode the input features for all compared baselines. We preserve the 10 most recent historical interaction records to construct the state representation. For sequences shorter than 10 interactions, we pad them with a padding token. The embedding dimensions for both state and goal are set to 64, and the batch size is fixed at 256. We utilize the Adam optimizer for all models, tuning the learning rate within the range of [0.0001, 0.0005, 0.001, 0.005]. Additionally, for methods that necessitate manual assignment of weights, we fine-tune these weights in the range of [0.1, 0.2, ..., 0.9] based on their performance on the validation set. The sample size $K$ in Algorithm 2 is set to 20 in the experiments. All experiments are conducted five times, each with different random seeds, and we report the mean and standard deviation of the results. When comparing the time complexity, each experiment was conducted with a separate NVIDIA RTX 3090 and AMD 3960X.

### B.5 COMPARISON BETWEEN MOGCSL AND MOPRLs

Figure 2 shows the performance comparison of MOGCSL to MOPRLs with different weight combinations.

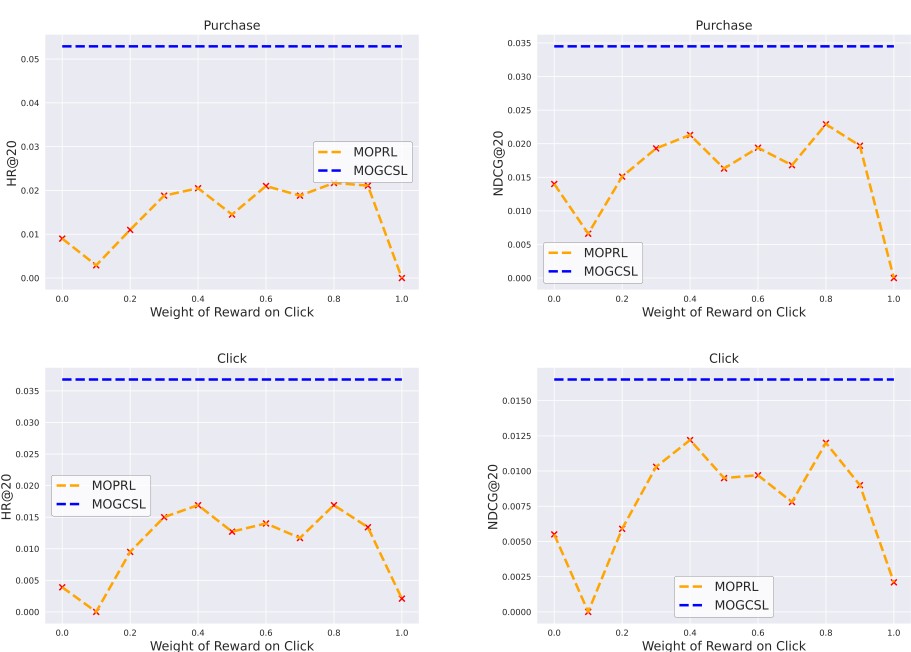

Figure 2: Comparison between MOGCSL and MOPRLs with different weight combinations on RetailRocket. Performance of MOGCSL is not dependent on the weights.

### B.6 COMPARISON OF MOGCSL-S W.R.T FACTORS

The details results of the performance comparison of MOGCSL-S with different factors for inference goal is shown in Figure 3.

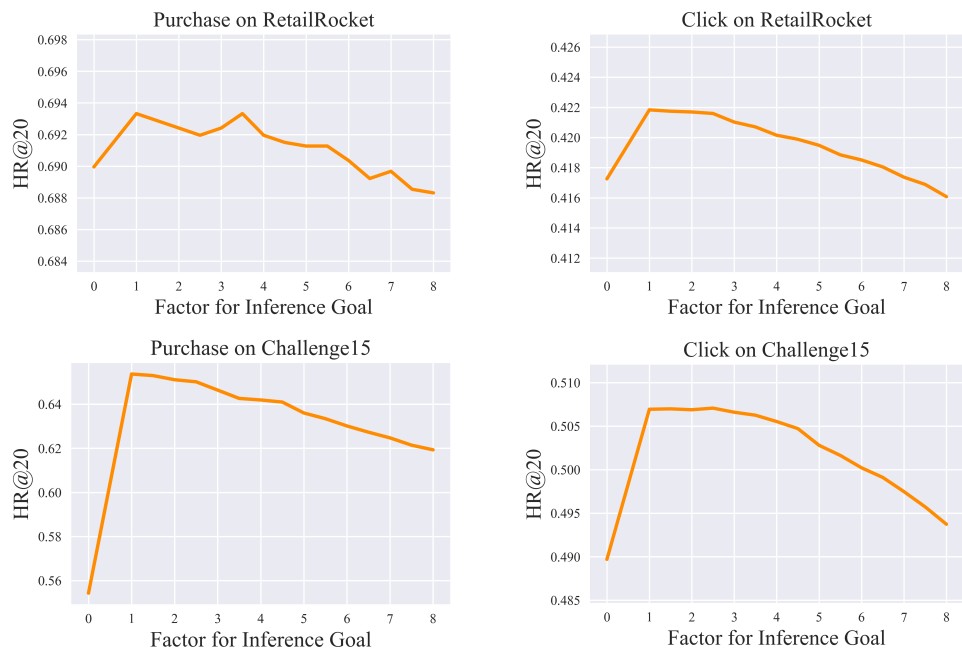

Figure 3: Performance of MOGCSL-S with different factors for inference goal.

## C ADDITIONAL DISCUSSIONS

### C.1 DISCUSSION ON COMPLEXITY

We have provided a detailed comparison of the space complexity and training cost of MOGCSL relative to other baselines, demonstrating its advantages in terms of smaller model size and faster training speed. Here, we give a further examination of the computational complexity of MOGCSL at inference time.

As outlined in Section 3.2, MOGCSL employs a simple policy network architecture built from basic attention modules combined with simple fully connected layers. Consequently, inference through the policy network incurs minimal computational cost, as further evidenced by its smaller model size compared to architectures such as MMOE or reinforcement learning frameworks (see Table 2). For the CVAE-based goal-selection module, we also adopt a lightweight MLP architecture with approximately 1.6M parameters, which converges rapidly during training (within 6 epochs). We've compared the training time of MOGCSL with the CVAE module (MOGCSL-C) with other baselines. As demonstrated in Table 5, we can see that even with the additional computation introduced by the CVAE, MOGCSL-C remains among the most efficient multi-objective learning methods.

Table 5: Comparison of time and space complexity of MOGCSL-C on RetailRocket.

|  | Model Size | Training time |
| --- | --- | --- |
| Share-Fix | 14.0M | 9.6Ks |
| Share-DWA | 14.0M | 5.3Ks |
| Share-PE | 14.0M | 5.6Ks |
| Share-Nash-MTL | 14.0M | 8.6Ks |
| Share-FAMO | 14.0M | 5.1Ks |
| MMOE-Fix | 14.1M | 10.2Ks |
| MMOE-DWA | 14.1M | 9.5Ks |
| MMOE-PE | 14.1M | 60.5Ks |
| MMOE-Nash-MTL | 14.1M | 12.4Ks |
| MMOE-FAMO | 14.1M | 8.8Ks |
| RMTL | 17.5M | 100.2Ks |
| PMORS | 14.2M | 10.4Ks |
| MOPRL | 9.1M | 3.2Ks |
| MOGCSL-C | 10.7M | 4.8Ks |

At inference, forward propagation through this CVAE introduces only negligible overhead due to the simplicity of the MLP structure. Although goal sampling introduces additional computation, it is required only once at the beginning of each inference sequence in practical applications. Subsequent

goals are computed efficiently through iterative subtraction of current rewards, as established in our theoretical analysis.

One important advantage of MOGCSL lies in its scalability to higher-dimensional objectives. By representing multiple objectives as a single vector input, MOGCSL naturally integrates additional goals without increasing architectural complexity. This vectorized formulation enhances scalability relative to conventional multi-objective approaches, which typically experience rapidly growing complexity with dimensionality. Importantly, this design eliminates the need for architectural modifications or substantial computational overhead when scaling to higher dimensions, as demonstrated in Table 2.

## C.2 DISCUSSION ON LEARNING PARADIGM

MOGCSL is designed for automated and effective learning in multi-objective settings by leveraging offline trajectories conditioned on successfully achieved multi-dimensional goals. The key intuition is to reframe the challenge of balancing potentially conflicting objectives into the more tractable task of learning to achieve realistic goals directly from observed trajectories. During training, MOGCSL directly learns to reach certain goals across multiple objectives by utilizing each trajectory in the training data as a demonstration of successful achievement of the goal that it actually achieves.

At inference, MOGCSL models the distribution of achievable goals across multiple objectives conditioned on the initial state. From this distribution, it samples goals that are both realistic and aligned with practical business requirements, as specified through a customizable utility function. By always grounding goal selection in attainable outcomes, MOGCSL naturally respects inter-objective trade-offs and avoids the need for manually imposed constraints or heuristic weight tuning. This design yields a more robust and flexible framework for multi-objective optimization, readily adaptable to diverse application scenarios.

## C.3 DISCUSSION ON GENERALIZABILITY

From the experiments and discussions in Section 4.4, we observed that different goal-choosing strategies perform differently on datasets with varying reward distributions. However, we emphasize our finding that simpler, statistics-based goal-selection strategies already outperform existing baselines across diverse scenarios, demonstrating MOGCSL's practical applicability especially where lower latency or reduced inference complexity is desired. The advanced CVAE-based algorithm provides additional performance gains specifically on datasets with higher goal values (as shown in Appendix B.8), offering practitioners the flexibility to choose strategies based on dataset characteristics and performance-complexity requirements. Rather than limiting generalizability, this observation provides valuable insights into appropriate strategy selection tailored to specific use cases in practice.

## C.4 DISCUSSION ON CONTRIBUTION AND NOVELTY

To further clarify the contribution and impact of our research, we summarize several important innovations that distinctly address critical challenges in general goal-conditional supervised learning and multi-objective recommendation scenarios:

- We are the first to integrate an MOMDP approach into GCSL's framework, extending the goal from a scalar to a multidimensional vector. This extension is non-trivial since defining and selecting achievable "high" multi-dimensional goals is conceptually complex and previously unexplored. Our paper proposes and evaluates both principled and heuristic goal-selection approaches, demonstrating their effectiveness across different datasets.
- We provide the first theoretical characterization of achievable goal distributions within the multi-objective setting of GCSL, formalized through Theorem 1 with complete proof (Section 3.3 and Appendix A).
- Building upon this theoretical foundation, we developed a novel goal-choosing algorithm that models the distribution of achievable goals over interaction sequences to select desirable high goals for inference. This algorithm is novel even for the single-objective case and addresses a fundamental challenge in GCSL applications.
- We were the first to demonstrate GCSL's capability to mitigate the harmful effects of noisy instances, which are common in real-world recommendation data. We showed that MOGCSL can leverage

the future impact of current actions across multiple objectives to achieve this ability (Section 3.4 and Appendix B.2).

### C.5 DISCUSSION ON GENERATIVE MODELS

As illustrated in Section 3.3, we chose CVAE for the goal-selection algorithm primarily because of its simplicity, efficiency, and stable training dynamics, which align closely with our objective of developing scalable methods suitable for real-world recommendation scenarios. Apart from that, we've conducted some preliminary experiments with diffusion-based methods such as DDPO, and the results showed no significant performance improvement over CVAE. Additionally, CVAE demonstrated consistently stable training dynamics and faster inference, essential attributes for practical deployments.

### C.6 DISCUSSION ON DATA ROBUSTNESS

As illustrated in Section 3.2, MOGCSL assumes the availability of reward signals within sequential data. In recommender systems, such reward signals typically derive naturally and easily from explicit user feedback, such as clicks, purchases, or engagement durations.

However, clear rewards might not be readily available in some other scenarios. For these situations, we propose the following discussions:

- Proxy rewards: Intermediate signals often effectively serve as proxies. For instance, browsing duration, add-to-cart events, or return visits in e-commerce scenarios can reliably approximate user satisfaction when explicit purchase data is sparse.
- Synthetic rewards: In cases like path planning or robotic control, synthetic metrics such as proximity to target states or successful task completion checkpoints can be constructed to substitute explicit rewards.
- Reward inference: Techniques such as inverse reinforcement learning or reward inference from demonstration could integrate seamlessly into MOGCSL, enabling application in less structured environments.

Furthermore, as discussed in Section 3.4, one of the MOGCSL's distinct advantages lies in leveraging long-term goals to effectively handle potentially noisy immediate reward signals. Our experiments in Appendix B.2 demonstrate that this mechanism significantly enhances robustness, enabling effective learning even when datasets contain ambiguous or imprecise signals.

### C.7 DISCUSSION ON LIMITATIONS AND FUTURE WORKS

Although MOGCSL demonstrates strong effectiveness and efficiency for multi-objective recommendation, we identify some potential limitations and corresponding future works.

First, the most prominent gains from the CVAE-based goal-selection strategy are observed on datasets with relatively high average goals. Simpler statistics-based strategies remain effective when average goals are low and offer practitioners flexibility in trading off performance against computational complexity. Nonetheless, further improving the CVAE-based goal-selection mechanism is a valuable direction. Potential approaches include data augmentation techniques (e.g., re-/down-sampling, active learning) to alleviate sparsity in the high-goal region. In addition, extending MOGCSL to an online training paradigm is a promising way to further mitigate this issue if an online environment is accessible: one can pre-train on offline data, deploy the policy to collect new trajectories with higher achieved goals, and iteratively retrain on this enriched dataset.

Second, our current experiments focus on using long-term multi-objective outcomes to downweight noisy instantaneous behaviors that are weak indicators of positive preference. A natural extension is to directly model and leverage explicit negative signals. This could be realized by encoding negative feedback (e.g., quits, skips, dislikes) as additional dimensions in the goal vector, and then incorporating penalties in the utility function to prioritize minimizing those signals during inference.

Third, although our evaluation uses real-world industrial recommendation datasets, we don't conduct online A/B testing with live users due to resource limitations. Industrial deployment and online

experimentation would provide additional evidence of practical impact and constitute an important direction, especially from an industry perspective.

Fourth, the theoretical analysis in this paper centers on formalizing the achievable goal distribution within the multi-objective GCSL framework, which constitutes a novel conceptual and methodological advancement for MOGCSL as well as the general GCSL problem. Nonetheless, additional formal analysis, such as theoretical comparisons with other multi-objective learning methods, could further clarify the benefits and limitations of the proposed framework.

Finally, while our study is motivated by empirical challenges in large-scale recommender systems (e.g., efficiency constraints and noisy interaction data), the MOGCSL formulation e broadly applicable to other multi-objective learning tasks. By reframing multi-objective optimization as learning to reach achievable multi-dimensional goals, the framework can potentially be applied to other domains, such as medical prediction, robotic control, and marketing optimization.

## D  LLM USAGE

Throughout the preparation of this paper, LLMs were used exclusively for improving clarity of expression and correcting typographical or grammatical errors. No other substantive assistance was employed.

