# OpenReview forum: "Goal-Conditioned Supervised Learning for Multi-Objective Recommendation"
_ICLR.cc/2026/Conference — Submitted to ICLR 2026_

### Official Review · Reviewer_2q32 · 2025-10-28

**Soundness:** 1
**Presentation:** 1
**Contribution:** 1
**Rating:** 2
**Confidence:** 2

**Summary:**

The paper presents an innovative approach to improving multi-objective recommendation systems by extending the Goal-Conditioned Supervised Learning (GCSL) framework to a multi-dimensional goal space. Instead of addressing a single scalar objective, the authors define multi-dimensional goals and develop a mechanism that enables the system to learn a recommendation policy that balances between different objectives according to preferences or contextual conditions.

**Strengths:**

While I believe the paper is still premature, I find the topic and the core idea very interesting.

**Weaknesses:**

In my opinion, the paper is written in an unfocused manner, which made it difficult for me to understand.

**Questions:**

Please address the weaknesses.

---

> ### Author Response · Authors · 2025-11-21
>
> Dear Reviewer 2q32,
>
>
>
> Thank you for your positive feedback on the significance and core research idea of our work. In our rebuttal, we would like to clarify the main content and contributions of our research. We hope the explanations below help provide a clearer understanding of our work, and we are always happy to address any further questions. We would also like to note that several other reviewers, such as Reviewer h32w and Reviewer 2BfV, provided positive feedback on the clarity, structure, and readability of the manuscript. As a result, we respectfully encourage the reviewer to consider revisiting the paper.
>
>
>
>
> Especially, as discussed in the Introduction, our research is motivated by the need for a multi-objective learning framework that is both efficient and robust to noisy data in industrial recommender systems. To this end, we propose Multi-Objective Goal-Conditioned Supervised Learning (MOGCSL) by extending GCSL to a multi-dimensional goal setting. The core idea is to transform the complex issue of training with explicit regulation among multiple objectives into the more manageable task of learning to achieve realistic multi-dimensional goals, as described in Section 3.1 and Appendix B.11. Through extensive experiments on real-world recommendation datasets, we demonstrate that MOGCSL achieves both superior performance and high efficiency.
>
> In summary, our main contributions are as follows:
>
>
> * We are the first to integrate an MOMDP approach into GCSL's framework, extending the goal from a scalar to a multidimensional vector. This extension is non-trivial since defining and selecting achievable "high" multi-dimensional goals is conceptually complex and previously unexplored. Our paper proposes and evaluates both principled and heuristic goal-selection approaches, demonstrating their effectiveness across different datasets.
>
> * We provide the first theoretical characterization of achievable goal distributions within the multi-objective setting of GCSL, formalized through Theorem 1 with complete proof (Section 3.3 and Appendix A).
>
> * Building upon this theoretical foundation, we developed a novel goal-choosing algorithm that models the distribution of achievable goals over interaction sequences to select desirable high goals for inference. This algorithm is novel even for the single-objective case and addresses a fundamental challenge in GCSL applications.
>
> * We were the first to demonstrate GCSL's capability to mitigate the harmful effects of noisy instances, which are common in real-world recommendation data. We showed that MOGCSL can leverage the future impact of current actions across multiple objectives to achieve this ability (Section 3.4 and Appendix B.2).

---

### Official Review · Reviewer_2BfV · 2025-10-31

**Soundness:** 3
**Presentation:** 4
**Contribution:** 3
**Rating:** 6
**Confidence:** 4

**Summary:**

This paper proposes MOGCSL, a framework that extends GCSL to handle multiple objectives by redefining goals as multi-dimensional vectors.  When applied to large-scale recommender systems, MOGCSL demonstrates strong scalability and robustness to noise. The experiment on recommender systems shows that MOGCSL outperforms many previous baselines and also outperforms MOPRL with different weights.

**Strengths:**

1. The experiment of this paper is comprehensive. They also show some comparisons about MOPRL and MOGCSL in the Appendix. I like the comparison between MOPRL and MOGCSL with different weights (MOGCSL remains unchanged).  This paper also contains many baselines and shows that MOGCSL achieves the best performance among many baselines.



2. The algorithms are proposed in a clear way. The paper is also well-structured and easy to follow.

3. The idea of using multi-objective GCSL to provide a solution for multi-objective learning is novel and interesting, which avoids knowing the weights of the objectives in advance. It is also reasonable to use VAE to generate goals.

**Weaknesses:**

1. In Line 146, the author claims that the dataset contains the $s_t$, which is the representation of the user's preferences. It does not practical in the real world. What does the real representation look like in your experiment?

2. The model structure introduced in Section 3.2 appears very similar to PRL, with the only difference being the addition of an extra MLP layer. Therefore, the improved performance over previous Shared-Bottom and MMoE models may largely stem from this architectural change (since MOPRL also achieves a good result), which weakens the claimed contribution of the paper.

3. Theorem 1, which serves as the main theoretical contribution of this paper, is trivial. It just claims that the expected goal is determined by some initial variable and the policy $\pi$.  This makes the theoretical contribution of this paper limited.

**Questions:**

1. It looks like these two objectives are correlated. What about the setting in which the objectives are in conflict?  Are there any benchmark tasks involving more conflicting objectives, beyond those focused only on recommender systems? Do the authors think MOGCSL will be general enough and work well in this setting?


2. In Line 427 and Table 2, the authors claim that their approach only requires an additional MLP layer and could be very efficient. Does the time reported in Table 2 include the training time for the VAE? Since MOGCSL additionally trains two CMAEs for goal generation, could the authors clarify whether the method remains efficient when accounting for this extra training cost?

3. In Section 4.4, the authors say that sparsity of training data within the high-goal space may contribute to the suboptimal performance of more advanced goal-choosing methods. Could you alleviate this through online data collection? The algorithm in this paper is purely offline. Is it possible to contain an online part in the algorithm?

4. In Line 276, how is the partial ordering obtained in practice? It seems to require weights across objectives in advance. In the experiments, no predefined partial ordering is used. Do the authors believe the method would perform better if such prior information were incorporated?

---

> ### Author Response · Authors · 2025-11-21
>
> Dear Reviewer 2BfV,
>
> We sincerely appreciate the reviewer’s careful review and positive feedback on the motivation, clarity, novelty, quality, and significance of our work. Your thoughtful and constructive comments will help us to further clarify our methods and innovations. We hope that our response below can be helpful to address your concerns.
>
>
> **W1. State Representation**
>
>
> Thank you for raising this point. We would like to clarify that in our formulation, the state is defined as the "representation of a user’s preferences at a given timestep", as stated in the original submission in Line 149.
> We use $s_t$ to denote the state, representing the user’s preference presentation at time $t$ within a trajectory, following the standard Markov Decision Process formulation.
> As detailed in Section 3.2 and shown in Figure 1, **the exact state embedding is constructed as shown in Eq. (2) (Line 204)**, by concatenating the sequential encoding of the user’s past behaviors, the timestep embedding, and the goal embedding. And this implementation precisely defines how the state representation is realized in our experiments.
>
>
> **W2. Model Structure**
>
>
>
>
> Thank you for highlighting this valuable point. We would like to clarify that our experimental design already accounts for this point for a fair model comparison. As described in Appendix B.4, all baselines, including our proposed method, employ **the same Transformer encoder with self-attention blocks** (introduced in Section 3.2) as the base module for encoding sequential features. The structural difference between MOGCSL, Shared-Bottom, and MMoE lies only in the top layers. As illustrated in Section 4.3, both MMoE and Shared-Bottom require separate towers for each objective, which leads to a substantial increase in model parameters as the number of tasks grows, while MMoE further introduces additional expert modules with gating mechanisms. In contrast, MOGCSL employs only a simple MLP layer above the shared bottom module. Despite this more concise and lightweight design, MOGCSL outperforms the other baselines in most metrics. Therefore, we believe the comparison is fair, and our claim of achieving superior performance with a simpler model structure remains well supported. Nevertheless, we have updated the manuscript to explicitly address and clarify this comparison setting in the main paper for clearer presentation.
>
> **W3. Theoretical Analysis**
>
>
>
> Thank you for raising this theoretical point. We'd like to argue that, although the conclusion of our theorem may appear simple and straightforward at first glance, it plays a crucial role in addressing the core question of “what are the general characteristics of achievable goals during inference?” To the best of our knowledge, our work is **the first to explicitly provide this key theoretical foundation for the achievable goal distribution** for the inference goal-selection mechanism in GCSL.
>
> Importantly, our proposed goal-selection algorithm is directly derived from this theorem, making it not only a theoretical contribution but also a practical backbone of our framework. Therefore, while the conclusion may seem intuitive, we regard this analysis as a significant contribution, both for our work and for advancing the broader research on GCSL and its extensions.

---

> ### Author Response · Authors · 2025-11-21
>
> **Q1. Objective Conflict**
>
>
>
> Thank you for this thoughtful comment. We'd like to clarify that the two objectives in our experiments, click and purchase, indeed exhibit a degree of conflict. As shown in Tables 1–2, several baselines (e.g., Share/MMOE-PE, Share/MMOE-FAMO) achieve comparable or slightly better click performance (like 1–2%) but at the cost of a substantial drop (like 11–20%) in purchase metrics relative to our MOGCSL. This reflects a key limitation of many existing multi-objective learning methods, such as PE and FAMO, which assume Pareto efficiency, requiring that improving one objective should not degrade performance on any other objective. These results demonstrate that optimizing for clicks by these methods actually severely hinders learning on purchase, confirming the inherent conflict between these objectives.
>
> However, as discussed in Section 4.2, this Pareto efficiency assumption usually doesn't align with real-world recommender systems, where improving high-value outcomes (e.g., purchases) with minimal loss in less critical metrics (e.g., clicks) is typically preferred. To address this, MOGCSL is designed to transform the complex issue of balancing conflicting objectives into the more manageable task of learning to **reach achievable multi-dimensional goals**, as described in Section 3.1 and Appendix B.11.
> In particular, MOGCSL is trained to learn this goal-reaching pattern on offline trajectories, each representing a successful demonstration of achieving certain multi-dimensional objectives that it actually achieves.
> At inference time, by assigning **achievable goals** through either a principled CVAE-based algorithm or simpler heuristic-based statistical methods, MOGCSL inherently respects objective trade-offs. MOGCSL then incorporates a customizable utility function to allow explicit prioritization according to specific business requirements from these achievable goals at inference time.
>
>
>
> This design allows MOGCSL to handle conflicts through goal attainability at inference, rather than relying on fixed optimization constraints or manual weight tuning for training, resulting in a more flexible and robust framework. As such, while this paper focuses on recommender systems, MOGCSL’s formulation can be broadly applicable to other multi-objective learning tasks, such as medical prediction, robotic control, and marketing optimization. In fact, we have conducted some preliminary experiments in molecular property prediction using the QM-9 dataset [1], comparing MOGCSL against the recent FAMO [2] baseline. These early results show promising performance improvements (average 12.3% gain across four molecular properties), demonstrating MOGCSL's potential versatility. These results were omitted from this submission to maintain focus on recommender systems. Nevertheless, we've now incorporated relevant discussions in the  Limitations and Future Works section (Appendix B.14) of the revised manuscript.
>
>
>
>
>
> **Q2. Complexity Analysis**
>
>
>
> Thank you for raising this valuable point. We'd like to clarify that, as noted at the beginning of Section 4.2, all experiments presented prior to Section 4.4 were conducted using the statistics-based goal-selection strategy. Accordingly, the complexity comparison in Table 2 is based on this same setup. The **complexity analysis of the CVAE modules is already provided in Appendix B.10**, and we have now extended this section to include more direct and quantitative comparisons. As shown in the updated results (Table 6), since the CVAE module is also implemented by a lightweight MLP architecture, MOGCSL with CVAE remains among the most efficient multi-objective learning methods.
>
> Furthermore, as discussed in Section 4.4, we'd like to address that MOGCSL with the simpler statistics-based strategies already achieves strong performance while maintaining excellent computational efficiency, making it particularly suitable for low-latency or resource-constrained scenarios. The CVAE-based MOGCSL offers further performance gains on datasets with higher goal values, providing practitioners with the flexibility to select the appropriate balance between performance and complexity based on application requirements. Overall, these results highlight the practical adaptability and efficiency of MOGCSL across different use cases.

---

> > ### Author Response · Authors · 2025-11-21
> >
> > **Q3. Online Adaption**
> >
> >
> >
> >
> > We appreciate the reviewer’s thoughtful and valuable comment. We fully agree that extending MOGCSL to an online training paradigm represents a promising direction to mitigate the data sparsity problem in the high-goal space. Conceptually, the policy could first be trained on existing offline data until convergence, and then deployed in an online environment to collect new trajectories. Since the learned policy already tends to achieve higher goals, the newly gathered data would naturally contain a greater proportion of high-goal samples. Retraining MOGCSL on this enriched dataset in an iterative manner could further enhance its performance and progressively alleviate data sparsity.
> >
> > However, as our work is motivated by and applied to industrial recommender systems, academic researchers like us typically lack direct access to deploy and evaluate models in real online environments with live users. Therefore, we tend to keep the offline training and evaluation setup in this study and compare our approach with offline multi-objective learning baselines. Nonetheless, we recognize the promising potential of online adaptation and have included a discussion of this direction in the Limitations and Future Works section (Appendix B.14) of the revised manuscript.
> >
> >
> >
> > **Q4. Utility Principle**
> >
> >
> >
> > As described in Section 4.4, the utility principle used in our experiments is defined in Eq. (8), which selects goals in a moderate manner such that no other achievable goal dominates it across all objectives. This design aligns with our aim of producing balanced performance across multiple objectives in the experiments.
> >
> > However, as discussed in Section 3.3, the utility principle in MOGCSL is flexible and can be adapted to prioritize different objectives based on application needs. For instance, if one objective holds greater business importance, practitioners can modify the partial ordering relationship accordingly. One of the simplest and most direct ways could be assigning higher weights to emphasize it. However, we note that this weighting mechanism differs fundamentally from loss weighting in training-based methods (e.g., MOPRL and DWA in our baselines). In MOGCSL, training is not guided by predefined objective weights; instead, the model learns policies to achieve diverse goals across all objectives demonstrated by offline trajectories. The relative importance of objectives is only applied during inference for goal selection among achievable goals, allowing the policy to flexibly align with empirical requirements.
> > This inherent flexibility enables practitioners to tailor MOGCSL to diverse real-world scenarios and evolving objective priorities with minimal retraining effort.
> >
> >
> > ---
> >
> > References
> >
> > [1] Lin, Xiao, et al. "A pareto-efficient algorithm for multiple objective optimization in e-commerce recommendation." RecSys. 2019.
> >
> > [2] Liu, Bo, et al. "Famo: Fast adaptive multitask optimization." NeurIPS. 2023.
> >
> > [3] Blum, Lorenz C., and Jean-Louis Reymond. "970 million druglike small molecules for virtual screening in the chemical universe database GDB-13." Journal of the American Chemical Society 131.25 (2009): 8732-8733.

---

### Official Review · Reviewer_h32W · 2025-11-01

**Soundness:** 2
**Presentation:** 2
**Contribution:** 2
**Rating:** 2
**Confidence:** 3

**Summary:**

This paper proposes MOGCSL, which extends GCSL to MTL by extending scalar goals to vectorized goals. A goal-selection algorithm is devised to identify achievable goals for inference. Experiments are performed on sequential recommendation task, using public datasets to showcase the efficacy of the proposed method.

**Strengths:**

1. The writing is clear and easy to follow.

2. The investigated multi-task learning problem is essential in recommendation system field.

**Weaknesses:**

1. In experiments, it would be beneficial to include more recent baselines with high reputations. On the MTL aspect, it seems that only DWA, PE and FAMO are involved. However, there are massive important MTL optimizers, including but not limited to MGDA, PCGrad,  UPGrad, MoDo, Nash-MTL, GradNorm, etc. Moreover, since authors mainly consider the sequential recommendation problem, it is not very clear why sequential recommendation models (e.g., FMLP-Rec, DuoRec, Longer) are not involved as baselines in the main comparison table or a separate section for case study.

2. Authors could isolate the comparison with MTL methods from the comparison with canonical sequential recommendation models.

3. A theoretical analysis demonstrating the superiority of the proposed methods compared to the state-of-the-art MTL method is lacking, which is a critical aspect to evaluate theoretical technical quality.

4. Experiments on real-world industrial datasets and online A/B analysis are lacking, which are critical aspects to evaluate the empirical technical quality of research works in recommendation system (e.g., sequential recommendation) field.

5. The motivation and rationale to select sequential recommendation as a testbed is not very persuasive. The problem aimed to solve by the proposed method seems to be very general instead of the unique problems in sequential recommendation.

6. This paper should discuss limitations and delineate future works to address these limitations in detail.

**Questions:**

Please see the weaknesses above.

---

> ### Author Response · Authors · 2025-11-21
>
> Dear Reviewer h32W,
>
> Thanks for your positive feedback on the motivation, clarity, and significance of our work.
> Your thoughtful and constructive comments will help us to further clarify our methods and innovations. We hope that our response below can be helpful to address your concerns.
>
>
>
>
> **W1. MTL baselines**
>
>
>
>
> We sincerely appreciate the reviewer’s suggestions regarding additional MTL baselines. However, we respectfully emphasize that our current baseline selection **already covers representative methods addressing multi-objective optimization** from various perspectives, spanning both architectural approaches (Shared-Bottom, MMOE) and optimization-based strategies (DWA, PE, FAMO).  For example, PE [1] was explicitly designed to provide Pareto-optimal recommendations, while FAMO [2] represents a recent and influential advancement in multi-objective scenarios, balancing efficacy and computational efficiency. The motivation behind FAMO closely aligns with our own goal of developing an efficient framework suitable for industrial multi-objective applications, making it a natural and highly relevant baseline for comparison.
>
> In addition to these general MTL methods, we also benchmarked against **several recent and strong multi-objective recommendation baselines**. In particular, we evaluated variants of PRL (MORPL), RMTL [3], which leverages offline reinforcement learning for sequential recommendation, and PMORS [4], a recently proposed supervised multi-objective recommendation framework.
>
> Nonetheless, responding to the reviewer's suggestions and with additional time and computational resources available, we have now included Nash-MTL [5], another well-known MTL approach, as part of our experiments. As clearly shown in both the revised manuscript and the accompanying results table, our proposed approach consistently outperforms these SOTA MTL methods.
>
>
>
> |               | Purchase | Purchase | Purchase | Purchase | Click | Click   | Click | Click   |
> |---------------|----------|----------|----------|----------|-------|---------|-------|---------|
> |               |   HR@10  |  NDCG@10 |   HR@20  |  NDCG@20 | HR@20 | NDCG@20 | HR@20 | NDCG@20 |
> | Share-Nash-MTL|   62.37  |   48.97  |   66.92  |   50.83  | 34.05 |  23.25  | 38.72 |  23.64  |
> | MMOE-Nash-MTL |   63.21  |   50.31  |   67.74  |   52.17  | 34.96 |  24.79  | 40.48 |  25.86  |
> | MOGCSL        |   65.43  |   52.92  |   69.28  |   53.90  | 36.30 |  25.24  | 41.92 |  26.67  |

---

> ### Author Response · Authors · 2025-11-21
>
> **W1/W2/W5. Sequential Recommendation Rationale and Comparison**
>
>
>
>
>
> Thank you for raising this valuable point. We would first like to clarify that our paper focuses on multi-objective learning within a session-based recommendation setting, rather than on developing new architectures for modeling users’ sequential behaviors and preferences. As described in Appendix B.4, to ensure a fair comparison, we employ the same simple Transformer encoder with self-attention blocks (introduced in Section 3.2) as the base module for encoding sequential features **across all baselines**. This ensures that user sequential behavior information is captured consistently in a concise and computationally efficient manner, aligning with our core motivation of designing a lightweight, scalable, and efficient framework for multi-objective recommendation. Importantly, we already include several baselines such as MOPRL and RMTL, which are explicitly designed for multi-objective sequential recommendation, corresponding precisely to our problem setting. In contrast, methods like FMLP-Rec, DuoRec, and Longer, **are not designed for multi-objective recommendation** and therefore are not directly comparable to our proposed approach.
>
> Second, regarding the rationale for adopting a sequential recommendation setup, this choice is driven by both the practical motivation of our work and the intrinsic requirements of the GCSL framework. From a practical standpoint, as explained in the Introduction, our primary motivation is to design a concise and scalable model suitable for large-scale recommendation systems with much noisy data. MOGCSL achieves this target by maintaining a simple architecture and low computational complexity (as shown in Section 4.3) while benefiting from the denoising capability validated in Section 4.4 and Appendix B.2. Consequently, the session-based sequential recommendation scenario provides a natural and meaningful context for evaluating the effectiveness of our framework.
>
> From a methodological perspective, session-based recommendation aligns with the sequential learning nature of GCSL. As detailed in Section 3.1, GCSL learns a policy by generating sequential actions aimed at achieving a specified goal. In this setup, goals are defined as the outcomes of user interaction sequences—in our case, cumulative rewards in recommendation sessions. Therefore, the session-based sequential recommendation setting inherently fits the GCSL formulation, as it provides the **necessary sequential behavior data to define both the goals and the corresponding goal-achieving policy**.
>
> Nevertheless, as noted by the reviewer, MOGCSL can indeed be naturally extended to other domains involving sequential data. We have now added a discussion addressing this potential extension in the Limitations and Future Works section (Appendix B.14) of the revised manuscript.
>
>
>
>
>
>
> **W3. Theoretical Analysis**
>
>
>
> Thank you for raising this theoretical point. However, we'd like to address that our work primarily focuses on extending GCSL to the multi-objective learning setting, aiming to empirically achieve high efficiency with a smaller model size and faster training, along with strong denoising capabilities. As outlined in the Introduction section, these aims are directly motivated by key challenges faced by industrial recommender systems when deploying multi-objective learning methods at scale. Through extensive analysis and experiments, we demonstrate MOGCSL's benefits on both fronts.
>
> Accordingly, our theoretical contribution **centers on formalizing the achievable goal distribution within the multi-objective GCSL framework**. This formulation, presented and proven in Theorem 1, provides the theoretical foundation for the inference goal-selection mechanism in MOGCSL. Building upon this, we further propose the CVAE-based goal-selection algorithm, which constitutes an important and novel methodological advancement of our work.
>
> Given that our paper is motivated primarily by empirical challenges in large-scale industrial applications, and that we already provide a solid theoretical basis for the core methodology, we consider extensive theoretical exploration for comparisons with other MTL frameworks as a direction for future research. We have included additional discussion of this aspect in the Limitations and Future Work section (Appendix B.14) of the revised manuscript.

---

> ### Author Response · Authors · 2025-11-21
>
> **W4. Experimental Setting**
>
>
>
> Thank you for raising this point. We'd like to clarify that the two public datasets used in our main experiments, Challenge15 and RetailRocket, are **exactly real-world industrial datasets**, as detailed in Appendix B.2.1. Especially, they are both collected from online e-commerce platforms by recording users’ sequential behaviors during recommendation sessions. For example, Challenge15 is collected from a big retailer in Europe, which is accepting recommender system as a service from YOOCHOOSE, as described by the original publication [6].
> Furthermore, both datasets are widely recognized benchmarks in prior studies on multi-objective recommendation, as evidenced in recent prominent works [3, 7, 8]. The realistic user behavior sequences captured in these datasets, including multiple objectives such as clicks and purchases, make them particularly suitable for evaluating the effectiveness of multi-objective recommendation methods such as ours.
>
> Regarding online A/B testing, we acknowledge its value. However, as academic researchers, we typically **do not have access to online production environments** for deploying and evaluating recommender systems with real users. Nonetheless, we recognize this as a valuable direction for future work, particularly for industrial researchers. We have added a relevant discussion of this point in the Limitations and Future Work section (Appendix B.14) of the revised manuscript.
>
>
> **W6. Limitations and Future Works**
>
>
>
> We appreciate the reviewer’s valuable suggestion. As discussed in our responses above, we've now added a Limitations and Future Works section (Appendix B.14) in the revised manuscript. It includes, but is not limited to, discussions of MOGCSL’s potential extensions to other domains, further theoretical exploration and comparison, and practical evaluation considerations such as online A/B testing.
>
>
> ---
>
> References
>
> [1] Lin, Xiao, et al. "A pareto-efficient algorithm for multiple objective optimization in e-commerce recommendation." RecSys. 2019.
>
> [2] Liu, Bo, et al. "Famo: Fast adaptive multitask optimization." NeurIPS. 2023.
>
> [3] Liu, Ziru, et al. "Multi-task recommendations with reinforcement learning." Proceedings of the ACM web conference 2023. 2023.
>
> [4] Jin, Jipeng, et al. "Pareto-based multi-objective recommender system with forgetting curve." Proceedings of the 33rd ACM International Conference on Information and Knowledge Management. 2024.
>
> [5] Navon, Aviv, et al. "Multi-task learning as a bargaining game." arXiv preprint arXiv:2202.01017 (2022).
>
> [6] Ben-Shimon, David, et al. "Recsys challenge 2015 and the yoochoose dataset." Proceedings of the 9th ACM Conference on Recommender Systems. 2015.
>
> [7] Stamenkovic, Dusan, et al. "Choosing the best of both worlds: Diverse and novel recommendations through multi-objective reinforcement learning." Proceedings of the fifteenth ACM international conference on web search and data mining. 2022.
>
> [8] Zaizi, Fatima Ezzahra, et al. "A multi-objective optimization approach for session-based recommendation systems." Journal of Intelligent Information Systems (2025): 1-30.

---

### Official Review · Reviewer_jBsw · 2025-11-02

**Soundness:** 2
**Presentation:** 2
**Contribution:** 2
**Rating:** 4
**Confidence:** 4

**Summary:**

This manuscript proposes the MOGCSL framework, extending the GCSL paradigm to multi-objective recommendation. Its core involves redefining GCSL's scalar goal into a multi-dimensional vector, which it claims avoids the complexity and dynamic loss weighting of traditional MTL architectures. By conditioning on high-value goals, the method aims to "de-noise" and focus on high-quality interactions. The manuscript also introduces a companion CVAE algorithm for goal selection at inference time. Experiments on two real-world e-commerce datasets show that MOGCSL significantly outperforms a range of strong baseline models on purchase-related metrics.

**Strengths:**

1.	The manuscript reframes multi-objective recommendation as a goal-conditioned supervised learning problem. This approach avoids the need for complex multi-task architectures or explicit conflict-handling during optimization, which is an insightful contribution.
2.	MOGCSL is shown to be more efficient than baselines in both model size and training speed. This is a crucial advantage for large-scale industrial applications.
3.	The work commendably tackles the difficult problem of goal selection during inference, a key challenge for GCSL. It provides an analysis and a theoretically-motivated CVAE-based solution.

**Weaknesses:**

1.	The manuscript's main technical contribution, a CVAE-based goal selection algorithm (MOGCSL-C), fails to consistently outperform a simple statistical heuristic (MOGCSL-S) and performs worse on one dataset. This undermines the algorithm's practical value and questions the significance of its theoretical backing.
2.	The explanation for the CVAE method's failure—data sparsity—feels like a post-hoc justification. The supporting experiment is relegated to an appendix and uses a different dataset, suggesting the method's effectiveness is a fundamental limitation tied to data properties, which is not adequately addressed in the main manuscript.
3.	The model's strong performance on purchase metrics comes at the cost of mediocre performance on click metrics, a trade-off that is not sufficiently discussed. This suggests an implicit, unmanaged trade-off, potentially contradicting the claim of avoiding explicit optimization constraints.
4.	The claim of denoising capability is supported only by a simplistic synthetic experiment in the appendix. The noise model used is not representative of complex, real-world scenarios, making the evidence for this claim weak.

**Questions:**

1.	Given that the CVAE-based method can underperform the simpler statistical one, how do you justify its added complexity for practical deployment?
2.	Does the CVAE's failure on sparse-reward data mean MOGCSL is only effective for datasets with dense rewards? Have you considered methods like data augmentation to mitigate this limitation?
3.	What is the mechanism behind the observed trade-off favoring purchase metrics over click metrics? How does the model resolve conflicting goals (e.g., high click and high purchase), and does it implicitly learn to prioritize one objective based on the training data?
4.	Your definition of "noise" equates it with low long-term utility. However, a quick exit after a click could be a strong negative preference signal, not just noise. Can your model distinguish between random noise and valid negative feedback?

---

> ### Author Response · Authors · 2025-11-21
>
> Dear Reviewer jBsw,
>
>
>
> Thanks for your positive feedback on the motivation, significance, methodology, and contribution of our work.
> Your thoughtful and constructive comments will help us to further clarify our methods and innovations. We hope that our response below can be helpful to address your concerns.
>
>
> **W1/Q1. Performance and Significance of CVAE**
>
>
> Thank you for raising this valuable point. We acknowledge the reviewer’s observation regarding dataset-specific performance. However, we emphasize our finding that simpler, statistics-based goal-selection strategies **already outperform existing baselines** across diverse scenarios, demonstrating MOGCSL's practical applicability especially where lower latency or reduced inference complexity is desired. The advanced CVAE-based algorithm provides additional performance gains specifically on datasets characterized by higher goal values. As illustrated in Section 4.4, this **provides practitioners the flexibility to choose appropriate goal-selection methods** for MOGCSL based on the dataset characteristics and performance-complexity trade-offs required by their specific application. Thus, rather than limiting the practicality of MOGCSL, this observation provides valuable insights into appropriate strategy selection tailored to practical use cases.
>
> Moreover, we'd like to clarify that the CVAE-based goal-selection algorithm constitutes only a portion of our technical contributions. As detailed at the end of the Introduction, we are the first to integrate an MOMDP into the GCSL framework, transforming the goal from a scalar into a multidimensional vector. Based on this, we present the first theoretical characterization of achievable goal distributions within the multi-objective GCSL setting, formalized and  proven in Theorem 1. Additionally, we demonstrate for the first time GCSL’s capability to effectively mitigate harmful impacts caused by noisy instances, a frequent issue in real-world recommendation datasets.
>
> Thus, our core technical contribution lies in developing a novel multi-objective learning framework leveraging GCSL, addressing and modeling the achievability of goals at inference. Within this framework, we explore and validate **both principled (MOGCSL-C) and heuristic (MOGCSL-S) goal-selection methodologies to derive these achievable goals**. These methods demonstrate complementary strengths across diverse datasets and empirical conditions. For example, if efficiency is the primary concern, MOGCSL-S may be preferable. Conversely, if recommendation accuracy is prioritized over computational constraints and training data with relatively high goal values are available, MOGCSL-C could be a better choice. We've extended Section 4.4 for a more detailed discussion in the updated manuscript.
>
> Besides, we'd like to address that we've **already included the complexity analysis of the CVAE modules in Appendix B.10** in our original submission. The CVAE components are intentionally designed to be lightweight, and our results show that even with the additional computation they introduce, MOGCSL remains among the most efficient multi-objective learning methods.

---

> ### Author Response · Authors · 2025-11-21
>
> **W2/Q2. Discussion of CVAE**
>
>
>
>
> We appreciate this valuable point. Initially, due to space constraints, we had placed additional discussions and experiments in the Appendix. Given the increased page limit at the rebuttal stage, we have now integrated all relevant content directly into Section 4.4 of the main paper.
>
> To further elucidate the empirical performance of CVAE, we have expanded our discussion in Section 4.4 by incorporating relevant analysis from existing literature on GCSL. Specifically, as already illustrated in Section 4.4, several prior studies [1, 2, 3] have similarly reported that overly ambitious goal selection strategies can negatively impact performance. In a recent work, RVDT [4], the authors provide an initial attempt at offering a more formal and theoretically grounded analysis of this problem. They draw similar conclusions by leveraging relevant theory and conclusions from offline reinforcement learning literature, showing that insufficient high-quality trajectories in offline datasets result in poor performance when GCSL policies are set overly optimistic. Applying these insights to our scenario, although the CVAE-based algorithm can identify potentially high-value goals, the GCSL model may lack adequate training data to effectively learn for these ambitious targets. Consequently, a more moderate strategy, such as setting goals based on the average cumulative goals observed in offline data multiplied by a hyper-parameter factor, could be empirically more effective in cases where training data with high average goals is unavailable.
>
> Actually, we did explore a couple of standard data augmentation methods (like re/down sampling), but we found that they provide very limited benefits in addressing this inherent limitation of offline data for GCSL, analogous to observations in the offline reinforcement learning literature. As a result, we'd like to leave more exploration on this trend with GCSL for future work. Nevertheless, we've included relevant discussion in the Limitation and Future Work section (Appendix B.14) in the updated manuscript.
>
>
> Additionally, as outlined in our response to W1/Q1, the simpler statistic-based approach **(MOGCSL-S) already outperforms existing baselines**, which is advantageous due to its computational efficiency and suitability for low-latency scenarios. And CVAE-based MOGCSL method can provide additional performance improvements if recommendation accuracy is prioritized and training data containing higher goal values are available. These results equip practitioners with the flexibility to select the most appropriate goal-selection method, strengthening the applicability of MOGCSL.

---

> ### Author Response · Authors · 2025-11-21
>
> **W3/Q3. Performance Trade-off**
>
>
>
>
> We sincerely appreciate this insightful comment. First, we'd like to address that MOGCSL demonstrates substantial performance improvements in critical purchase metrics, achieving an increase of 11–20% on purchase metrics compared to the strongest baseline method on click like MMOE-PE, while only experiencing a minimal reduction of 1–2% on click metrics. These results and their implications are thoroughly presented and discussed in Section 4.1.
>
> To further clarify this phenomenon, we highlight a fundamental assumption with many existing multi-objective learning methods, including our baselines such as PE and FAMO. These approaches typically optimize for Pareto efficiency, which assumes that improving one objective should not degrade performance on any other objective. However, this assumption does not align well with the realities of recommender systems. In practice, a significant improvement in high-value objectives (such as purchase) accompanied by only minor reductions in less critical objectives (such as clicks) is often preferable in practical scenarios.
>
>
> In contrast, as detailed in Appendix B.11, MOGCSL is designed to transform the complex issue of regulating conflicting objectives through training stages into the more manageable **task of learning to reach achievable multi-objective goals**. Specifically, MOGCSL is trained to learn this goal-reaching pattern on offline trajectories, each representing a successful demonstration of achieving certain multi-dimensional objectives it actually achieved. At inference time, by assigning **achievable goals** through either a principled CVAE-based algorithm or simpler heuristic-based statistical strategy, MOGCSL inherently respects objective trade-offs. MOGCSL then incorporates a customizable utility function to allow explicit prioritization according to specific business requirements from these achievable goals.
> In our experiments, we implement this utility by Eq. (8), which selects goals in a moderate manner such that no other achievable goal dominates it across all objectives. This is significantly different from the Pareto efficiency criteria and allows more balanced performance across these two objectives. Overall, our framework manages conflicts through goal attainability at inference rather than relying on predefined optimization constraints or heuristic weight tuning for training, resulting in a more robust and flexible approach.
> Nevertheless, we've followed the reviewer's suggestions to include more detailed discussions in Section 4.2.
>
>
>
> **W4. Denoising Capability**
>
>
>
> Thank you for raising this valuable point. First, we'd like to clarify the motivation behind conducting noise-filtering experiments using synthetic datasets. Due to the **unavailability of labels distinguishing between random and genuinely preference-driven user interactions in real-world datasets**, we explicitly modeled this mechanism through carefully designed synthetic experiments, as detailed in Appendix B.2. Specifically, clicks associated with lower long-term rewards (i.e., goals) were intentionally accompanied by higher noise levels to simulate behaviors less reflective of users' actual preferences.
> In fact, several prior studies on recommender system denoising [5, 6] have highlighted that such noises (i.e., users' noisy instant behaviors that don't align with their long-term, real interests) are very common and widely exist in real-world recommendation datasets.
>
> By design, MOGCSL inherently prioritizes the learning from training samples with higher long-term goal values across multiple objectives. Coupled with its superior empirical performance on both synthetic and real-world industrial datasets compared, MOGCSL's inherent denoising capability is validated.
>
> Nevertheless, following the reviewer's suggestion, we have expanded our relevant discussion in Section 3.4 and Appendix B.2 of the revised manuscript.

---

> > ### Author Response · Authors · 2025-11-21
> >
> > **Q4. Noise Signal**
> >
> >
> >
> > Thank you for the thoughtful question. We'd like to clarify that noise in our work refers to instantaneous behaviors (e.g., individual clicks) that are weak or ambiguous indicators of users' preferences when measured against long-term utility.
> > A quick exit after a click can be one of the main reasons to lower this long-term utility and is thus treated as non-evidence for positive preference. But our framework does not model the cause explicitly; instead, it weights evidence by outcomes: when high goals (high long-term utility) are specified at inference, the policy relies on patterns learned from high-utility trajectories and discounts interactions associated with low utility.
> >
> > If distinguishing negative feedback explicitly is desired, MOGCSL readily supports it. We can first encode negative signals (e.g., quit, skips, dislikes) as an additional objective dimension in the goal vector, and then incorporate penalties in the utility function to prioritize minimizing those signals during inference. While our current experiments were not designed for this purpose, it represents a promising direction for future research. We have added relevant discussion in the Limitations and Future Work section (Appendix B.14) of the revised manuscript.
> >
> >
> > ---
> >
> > References
> >
> >
> > [1] Chen, Lili, et al. "Decision transformer: Reinforcement learning via sequence modeling." Advances in neural information processing systems 34 (2021): 15084-15097.
> >
> > [2] Xin, Xin, et al. "Rethinking reinforcement learning for recommendation: A prompt perspective." Proceedings of the 45th international ACM SIGIR conference on research and development in information retrieval. 2022.
> >
> > [3] Zheng, Qinqing, Amy Zhang, and Aditya Grover. "Online decision transformer." international conference on machine learning. PMLR, 2022.
> >
> > [4] Bai, Wensong, et al. "Rebalancing Return Coverage for Conditional Sequence Modeling in Offline Reinforcement Learning." The Thirty-ninth Annual Conference on Neural Information Processing Systems. 2025.
> >
> > [5] Zhang, Kaike, et al. "Robust recommender system: a survey and future directions." ACM Computing Surveys 58.1 (2025): 1-38.
> >
> > [6] Chen, Jiawei, et al. "Bias issues and solutions in recommender system: Tutorial on the RecSys 2021." Proceedings of the 15th ACM conference on recommender systems. 2021.

---

### Author Response · Authors · 2025-11-26

Dear Reviewers,

Thank you once again for the time and care you have dedicated to evaluating our paper and for the constructive feedback you have provided. With the reviewer–author discussion deadline (December 3) approaching **in the coming week**, we would like to confirm that we have adequately addressed all of your comments. If there are any additional questions, clarifications, or suggestions you would like us to consider, please feel free to let us know. Your insights are invaluable to us, and we are eager to address any remaining issues to further improve our work.


Best regards,

The Authors

---

### Author Response · Authors · 2025-12-02
**Rebuttal Summary**

Dear ACs and SACs,

Thank you for overseeing the review process and for all your efforts in evaluating our paper and rebuttals. We greatly appreciate your work. Below, we briefly summarize our key responses to the reviewers’ comments.

&nbsp;
## **Goal-choosing Algorithm and Performance**


We clarify that our core technical contribution lies in developing a novel multi-objective learning framework leveraging GCSL. To address and model the achievability of multi-dimensional goals, we **propose both principled (MOGCSL-C) and heuristic (MOGCSL-S) goal-selection strategies**. Our results show that **MOGCSL-S already outperforms existing baselines** across diverse scenarios, which is desirable especially when lower latency and complexity are needed. Meanwhile, the CVAE-based MOGCSL-C can deliver additional performance gains when higher accuracy is prioritized and sufficient training data with high-goal values are available. These findings offer practitioners flexibility in choosing the most suitable goal-selection strategy, enhancing rather than limiting the practicality of MOGCSL.

For the CVAE-based strategy, we've **already provided a detailed complexity analysis in Appendix B.10**, showing the high efficiency of the CVAE modules. We've also added expanded discussion and experimental comparisons of the different goal-selection strategies in Section 4.4.

 &nbsp;

## **Performance Trade-off and Objective Conflict**

First, we emphasize that MOGCSL delivers substantial gains on high-value purchase metrics, achieving 11–20\% improvements over the strongest click-based baseline, while incurring only a 1–2\% decrease on click metrics. This trade-off is highly desirable in practical recommender systems, where large improvements in high-impact objectives (such as purchases) paired with minimal reductions in lower-value objectives (such as clicks) are almost always the preferred outcome.

MOGCSL achieves this by **reframing the challenge of regulating conflicting objectives during training into the more tractable problem of learning to reach achievable multi-objective goals**. The model is trained to learn how to achieve the realistic multi-dimensional goals demonstrated by offline trajectories. At inference, MOGCSL first selects **achievable goals**, either via a principled CVAE-based method or a simpler statistics-based strategy, which **naturally enforces the inherent trade-offs among objectives**. MOGCSL then applies a customizable utility function to choose a desirable goal from achievable ones. In our experiments, we adopt the utility in Eq. (8), which selects goals in a moderate manner such that no other achievable goal dominates the chosen one across all objectives. This differs fundamentally from Pareto-efficiency–based or hard-regularization approaches (e.g., MMOE-PE), resulting in a more robust, flexible, and practically aligned multi-objective recommendation framework.



&nbsp;
## **Theoretical Analysis and Contribution**



We address that our work focuses on extending GCSL to the multi-objective learning setting by reframing the problem as the task of learning to reach achievable multi-objective goals. Correspondingly, our theoretical contribution centers on **formalizing the achievable goal distribution within the multi-objective GCSL framework**. This formulation, established in Theorem 1, provides the key theoretical foundation for MOGCSL’s inference-time goal-selection mechanism. Building on this insight, we further introduce a CVAE-based goal-selection algorithm, which serves as an important and novel methodological component of our approach.

This theorem directly addresses the core question in GCSL: “What are the general characteristics of achievable goals during inference?” To the best of our knowledge, our work is the first to explicitly formalize this achievable goal distribution for GCSL, thereby providing the theoretical foundation necessary for principled goal selection at inference time.

---

> ### Author Response · Authors · 2025-12-02
> **Rebuttal Summary**
>
> &nbsp;
> ## **Clarifications of Key Misunderstandings**
>
>
>
> We highlight and clarify several key misunderstandings reflected in the reviews. First, our paper **already includes a comprehensive set of representative multi-objective learning baselines** from various perspectives, including architectural approaches (Shared-Bottom [1], MMOE [1]), optimization-regulation methods (PE [2], DWA [3], Nash-MTL [4], FAMO [5]), and strong and most recent multi-objective recommendation baselines like RMTL [6] and PMORS [7]. Because our work specifically addresses multi-objective learning in a session-based recommendation setting, sequential recommendation methods designed for **single-objective optimization are not appropriate baselines for comparison**.
>
> Second, as clearly described in Sections 3.2 and 4.1, the datasets used in our experiments are **exactly real-world industrial datasets**, and they **do not contain any direct representation of user preferences**. Additionally, we note that expectations for online A/B tests **are not reasonable for academic researchers like us** without access to industrial production systems.
>
> Finally, we clarify that we've **already ensured a fair and consistent complexity comparison** across all baselines by keeping the sequential feature encoder architecture identical for every method, as detailed in Section 4.1 and Appendix B.4.
>
> &nbsp;
> ## **Denoise Capability**
>
>
> We clarify that the noise in our work refers to instantaneous user behaviors (e.g., individual clicks) that serve as weak or ambiguous signals of long-term user preference. Because real-world datasets do not provide labels distinguishing noisy interactions from genuine preference-driven actions, we conducted synthetic experiments to directly evaluate MOGCSL’s denoising ability. As illustrated in Section 3.4 and Appendix B.2, MOGCSL inherently prioritizes training samples with higher long-term goal values across multiple objectives, enabling it to naturally suppress the influence of noisy, instant behaviors. Combined with its strong empirical performance on both synthetic and real-world industrial datasets containing substantial amounts of such noisy data, this design validates MOGCSL’s inherent denoising capability.
>
> ---
>
>
> References
>
> [1] Ma, Jiaqi, et al. "Modeling task relationships in multi-task learning with multi-gate mixture-of-experts." Proceedings of the 24th ACM SIGKDD international conference on knowledge discovery & data mining. 2018.
>
> [2] Lin, Xiao, et al. "A pareto-efficient algorithm for multiple objective optimization in e-commerce recommendation." RecSys. 2019.
>
> [3] Liu, Shikun, Edward Johns, and Andrew J. Davison. "End-to-end multi-task learning with attention." Proceedings of the IEEE/CVF conference on computer vision and pattern recognition. 2019.
>
> [4] Navon, Aviv, et al. "Multi-task learning as a bargaining game." arXiv preprint arXiv:2202.01017 (2022).
>
> [5] Liu, Bo, et al. Liu, Bo, et al. "Famo: Fast adaptive multitask optimization." Advances in Neural Information Processing Systems 36 (2023): 57226-57243.
>
> [6] Liu, Ziru, et al. "Multi-task recommendations with reinforcement learning." Proceedings of the ACM web conference 2023. 2023.
>
> [7] Jin, Jipeng, et al. "Pareto-based multi-objective recommender system with forgetting curve." Proceedings of the 33rd ACM International Conference on Information and Knowledge Management. 2024.

---

### Meta-Review · Area_Chair_jasd · 2026-01-06

**Summary:**

This paper proposes the MOGCSL framework, extending goal-conditioned supervised learning to multi-objective recommendation, which drew four reviewers’ evaluations with divergent concerns. Reviewer jBsw questioned the practical value of MOGCSL-C, noting it underperformed the heuristic MOGCSL-S on some datasets, with post-hoc justifications for its failure unconvincing. They also pointed out the under-discussed performance trade-off favoring purchase metrics over clicks and weak evidence for the model’s denoising capability, as it relied on simplistic synthetic experiments. Reviewer h32W criticized the insufficient baselines, lacking both advanced multi-task learning optimizers and sequential recommendation models. They also highlighted the absence of in-depth theoretical analysis, online A/B tests, clear rationale for choosing sequential recommendation as the testbed, and detailed limitations and future work discussions.
Reviewer 2BfV argued the model structure was overly similar to PRL, suggesting performance gains might stem from architectural tweaks rather than core innovations, and deemed Theorem 1 theoretically trivial. They further probed MOGCSL’s generality in handling conflicting objectives, efficiency when accounting for CVAE training costs, feasibility of online data collection to alleviate data sparsity, and the role of partial ordering in goal selection. Finally, Reviewer 2q32 claims that the paper was unfocused and hard to follow, though this view conflicted with other reviewers’ feedback on presentation clarity. Although all the reviewers have not evolved in the author reviewer discussion, the intial score is far from the accept line. The limitations of method and theorm are not fully addressed.

**Reviewer Concerns:**

For Reviewer jBsw, the authors clarified the complementary strengths of MOGCSL-C and MOGCSL-S, justified the purchase-click trade-off’s practical rationality, and supplemented discussions on denoising capability with literature support. They also added complexity analysis for CVAE modules. Reviewer h32W’s demands were met by incorporating Nash-MTL as a new baseline, explaining why sequential recommendation was a suitable testbed, and adding a limitations section. Reviewer 2BfV’s queries were resolved via clarifying the fair model comparison setup, emphasizing Theorem 1’s pioneering value for achievable goal distribution, and elaborating on MOGCSL’s efficiency and generality. Reviewer 2q32’s vague critique of the paper’s unfocused writing was not fully addressed, and I recommend to add a figure to describe the method.

**Reviewer Scores:**

Reviewer 2q32 provides limited review and has not evolved in the discussion. The score may be not fair.

---

### Decision · Program_Chairs · 2026-01-26

Reject